# In situ cryo-electron tomography reveals the progressive biogenesis of basal bodies and cilia in mouse ependymal cells

Shanshan Ma [1,7], Luan Li [2,3,7], Zhixun Li[1,7], Shenjia Luo[1,7], Qi Liu [4,7], Wenjing Du[1], Benhua Qiu[2,3], Miao Gui [4] ✉, Xueliang Zhu [2,3,5] ✉ & Qiang Guo [1,6] ✉

Cilia, essential organelles for cell motility and signaling, comprise an axoneme extended from the basal body (BB). The assembly process of BBs and axonemes during ciliogenesis, however, remains largely unknown due to the lack of structural information. Here, we leverage in-situ cryo-electron tomography to capture within mouse ependymal cells the dynamic processes of BB biogenesis and multiciliogenesis at various stages. This approach enables 3D visualization of the complete motile machinery, revealing the continuous microtubule-based scaffold from BBs to axonemes at sub-nanometer resolution with unprecedented structural details. Furthermore, we elucidate along BBs and cilia heterogeneous landscapes of microtubule-binding proteins underlying the establishment of structural periodicity and diverse subregions. Notably, the chronological binding patterns of microtubule-inner proteins (e.g., CEP41) correlate with the progressive assembly of ciliary beating machinery. We also resolve a substructure that borders the BB and the axoneme. Our findings provide key insights into intricate orchestrations during ciliogenesis.

Cilia are slender, hair-like protrusions from the surfaces of a wide array of eukaryotic cells, which are divided into two main categories: immotile cilia and motile cilia[1,2]. Immotile cilia are recognized as cellular antennae that function as a signaling hub[2], whereas motile cilia can propel cell movements or luminal fluid flows which effectively clear debris in trachea and promote cerebrospinal fluid (CSF) circulation in brain ventricles[3]. Long cilia that drive the locomotion of sperm and some protozoa are termed flagella[1,4]. Cilia dysfunction in human results in a broad spectrum of genetic disorders known as ciliopathies[5], including developmental abnormalities, sensory organ degenerations, chronic respiratory problems, hydrocephalus and sterility.

Cilia are composed of axonemal microtubules elongated from basal bodies (BBs) and exterior specialized membrane[6]. The BB is a specialized centriole usually containing nine triplet microtubules (TMTs), with a barrel-shaped cylindrical structure of ~450 nm in height and 250 nm in outer diameter[7]. BBs template ciliogenesis via extracellular or intracellular pathway[8,9]. In the extracellular pathway, BBs migrate and dock to the apical surface of cell membrane, while in the intracellular pathway, BBs recruit vesicles at the distal appendages and

[1]State Key Laboratory of Membrane Biology, Center for Life Sciences, Academy for Advanced Interdisciplinary Studies, School of Life Sciences, Peking University, Beijing, China. [2]Key Laboratory of Multi-Cell Systems, Shanghai Institute of Biochemistry and Cell Biology, Center for Excellence in Molecular Cell Science, Chinese Academy of Sciences, Shanghai, China. [3]University of Chinese Academy of Sciences, Beijing, China. [4]Department of Obstetrics and Gynecology, Sir Run Run Shaw Hospital, School of Medicine and Liangzhu Laboratory, Zhejiang University, Hangzhou, China. [5]Key Laboratory of Systems Health Science of Zhejiang Province, School of Life Science, Hangzhou Institute for Advanced Study, University of Chinese Academy of Sciences, Hangzhou, China. [6]Changping Laboratory, Beijing, China. [7]These authors contributed equally: Shanshan Ma, Luan Li, Zhixun Li, Shenjia Luo, Qi Liu. ✉e-mail: miaogui@zju.edu.cn; xlzhu@sibcb.ac.cn; guo.qiang@pku.edu.cn

the fusion of ciliary vesicles leads to the formation of ciliary sheath. Distal to the BB is the axoneme, which, in motile cilia, canonically consists of nine peripheral doublet microtubules (DMTs) surrounding a central pair (CP) of singlet microtubules[3]. The transition zone (TZ), featured by Y-links and a ciliary necklace, links proximal axonemal DMTs with ciliary membrane and acts as a diffusion barrier that controls protein trafficking between the cytoplasm and cilia[10,11].

The advent of cryo-electron microscopy (cryo-EM) and cryo-electron tomography (cryo-ET) has revolutionized our understanding of DMTs by enabling high-resolution structural and compositional determinations from diverse tissues and species[12–18]. Despite the conserved exterior 96-nm repeat and interior 48-nm periodicity, DMTs of motile cilia exhibit remarkable compositional heterogeneity[12–14,19]. This variation is likely an adaptation that allows cilia to achieve diverse beating patterns and function effectively within a wide range of environments. However, limitations in sample preparation and image processing techniques have hindered high-resolution structural studies of the BB and TZ. Nevertheless, medium-resolution cryo-ET studies, mainly focusing on protists, have provided valuable insights into the BB architecture[20–25], comprising the cartwheel-containing and inner scaffold-containing regions from proximal to distal ends. The proximal cartwheel and A-C linker corporately define the ninefold symmetrical configuration of BBs[20,21,23]. Despite these advancements, the current structures are inadequate for the identification of differential microtubule-binding proteins within BBs compared to axonemes. Furthermore, apart from the rudimentary model of BB-axoneme transition proposed from motile cilia in bovine tracheae[26], the intricate details underlying the dynamic processes of BB biogenesis and ciliogenesis remain elusive.

Multiciliated cells, with up to hundreds of motile cilia on the apical side of plasma membrane[27], provide a notable advantage for addressing the issues. Multiciliogenesis in mammals requires efficient generation of numerous BBs via deuterosome- and centriole-dependent pathways, with deuterosome-dependent BB biogenesis being the most prominent[28–30]. Deuterosomes are electron-dense spherical organelles capable of de novo basal body biogenesis[31]. Deup1 (deuterosome protein 1), a paralogue of the mother centriole-localized protein CEP63, forms the core of deuterosomes[32,33]. The Deup1 core further recruits other proteins critical for centriole biogenesis such as CEP152 and PLK4[34–36], so that deuterosomes function as centriole-independent platforms for massive BB biogenesis through a mechanism similar to that of centriole-dependent centriole biogenesis[28–30,32,33]. Deuterosome sizes vary considerably in different cells and tissues, with larger ones being capable of harboring more procentrioles[29,31,37]. Ependymal cells, with longer cilia[38] and larger deuterosomes[29,33,39] than those in tracheal epithelial cells, were chosen to investigate BB biogenesis and multiciliogenesis in the study. The cells are terminally differentiated multiciliated cells lining the surface of cerebral ventricles and central canal of the spinal cord, playing a key role in driving unidirectional flow of CSF and maintaining brain homeostasis[40].

This study leverages an integrative approach, encompassing correlative light and electron microscopy (CLEM), cryo-focused ion beam (cryo-FIB) milling, cryo-ET, and subtomogram averaging (STA) to capture unprecedentedly detailed three-dimensional (3D) snapshots of BB biogenesis and ciliogenesis in cultured mouse ependymal cells (mEPCs). This approach facilitates visualization of the beating machinery at the nanoscale, including the BB and axoneme, as well as their definitive demarcation point. The decent resolution enables the identification of spatial distributions of microtubule-binding proteins within BBs, including multiple WD40 domain-containing proteins. Furthermore, the study reveals the chronological binding patterns of microtubule-inner proteins (MIPs), exemplified by CEP41, throughout distinct cilia maturation stages, which suggests a highly orchestrated process of ciliogenesis.

## Results

### Establishing a pipeline that captures various stages of ciliogenesis in mEPCs

The BB biogenesis in mEPCs is divided into six stages (I to VI) (Fig. 1a, b)[29,32], followed by multiciliogenesis, which initially occurs asynchronously in stage-VI cells (Fig. 1b)[33]. In the study, we used mEPCs cultured to day 3 post serum starvation, a timepoint capturing multiciliated cells at various differentiation stages (Fig. 1a, b)[33,40], for structural analysis. This heterogeneity, where BBs at distinct developmental phases coexist, makes the early-stage ependymal cells a powerful model for capturing structural snapshots of BB assembly. In comparison, multiciliated cells became more abundant at day 10 (Fig. 1b, c), whose cilia were generally longer (median = 9.7 µm at day 10 vs. 4.1 µm at day 3) and more uniform in length (Fig. 1b, d).

To guide the selection of regions of interest, we expressed GFP-Deup1 in mEPCs as a fluorescent marker (Supplementary Fig. 1a–d)[32,33]. To avoid possible premature deuterosome formation, a mouse Deup1 promoter was used to drive the expression (see section "Methods" for details)[33]. The integration of cryo-CLEM, cryo-FIB and cryo-ET enabled the study of asynchronous BB biogenesis and multiciliogenesis in mEPCs, where deuterosome-attached BBs and ongoing ciliogenesis coexisted (Fig. 1e–h, Supplementary Fig. 1e–g and Supplementary Movie 1). Additionally, our observations revealed that the distal end of certain BBs features a specialized annular structure composed of 27 rods, which will be discussed later (Fig. 1e–h and Supplementary Movie 1).

Deuterosomes appear as electron-dense spherical structures of varying sizes (Fig. 1e, f and Supplementary Fig. 1e–g) as reported[29,33,39]. The central hub (CH) of cartwheel directly stretches out from deuterosomes and extends beyond the proximal end of BBs (Fig. 1e, f and Supplementary Movie 1). This phenomenon aligns with the protruding cartwheel observed in procentrioles of human U2OS cells[41], reinforcing the notion that the cartwheel assembles earlier than TMTs of BBs or procentrioles[42]. And, the cartwheel is missing when BBs template ciliogenesis (Fig. 1g, h), in agreement with the absence of Sas6, a critical component of the cartwheel[43], in mouse tracheal epithelial cells (mTECs)[32] and mEPCs[33] at stage VI.

In summary, we have established an effective and practical pipeline for visualizing deuterosome-dependent BB biogenesis and ciliogenesis at various stages in mammalian cells.

### Variations along the central axis of the BB

By employing STA, we successfully resolved the structures of TMT and DMT (Supplementary Fig. 1h–k and Supplementary Table 1). Focused alignment on A and B tubules of BBs yielded a TMT$^{AB}$ structure at a resolution of 7.6 Å (Supplementary Fig. 1h, j and Supplementary Table 1). Subsequent classification revealed three TMT subsets distributed from proximal to distal along BBs, which are complete TMT with pinhead (referred to as TMT$^{PH}$), complete TMT with an inner scaffold (referred to as TMT$^{IS\text{-}comC}$), and incomplete TMT with an inner scaffold (referred to as TMT$^{IS\text{-}incomC}$) with overall resolutions of 9.1 Å, 8.8 Å and 9.2 Å, respectively (Fig. 2a and Supplementary Fig. 1h, j). Different from the known 96-nm arrangement of DMT[13], all observed microtubule-associated proteins (MAPs) of BBs exhibit an 8-nm periodicity (Fig. 2b, c), in agreement with the structure of BB in bovine trachea[26]. Following the annotations in sperm DMT[14], we introduced the names 'mouse basal body MIP (mBMIP)' and 'mouse basal body MAP (mBMAP)' for MIPs and MAPs defined in mouse BBs in the study. mBMAP1 fills the seam of A tubule along BBs to maintain its integrity and provides the attachment sites for A-C linker of TMT$^{PH}$ (Fig. 2c). Moreover, two helical densities fill the outer junctions of A-B tubules (referred to as OJ$^{AB}$) and B-C tubules (referred to as OJ$^{BC}$) to potentially strengthen the TMT stability (Fig. 2c).

Based on the differential longitudinal localization of the three TMT structures, BB was subdivided into three regions with distinct

structural features (Supplementary Fig. 2a, b). Particularly, TMT$^{PH}$ is strongly correlated with the cartwheel region (Supplementary Fig. 2b). In detail, TMT$^{PH}$, with the A-C linker and the L-shaped pinhead, constitutes the ninefold symmetric proximal end of the BB

(Fig. 2a and Supplementary Fig. 2a)[44,45]. The A-C linker tethers neighboring TMT$^{PH}$ by binding to both A08-A10 of one TMT and C08-C10 of the adjacent TMT (Fig. 2a and Supplementary Fig. 2a), like that in *Chlamydomonas*[21]. The pinhead, located at A03-A04 of TMT$^{PH}$, is

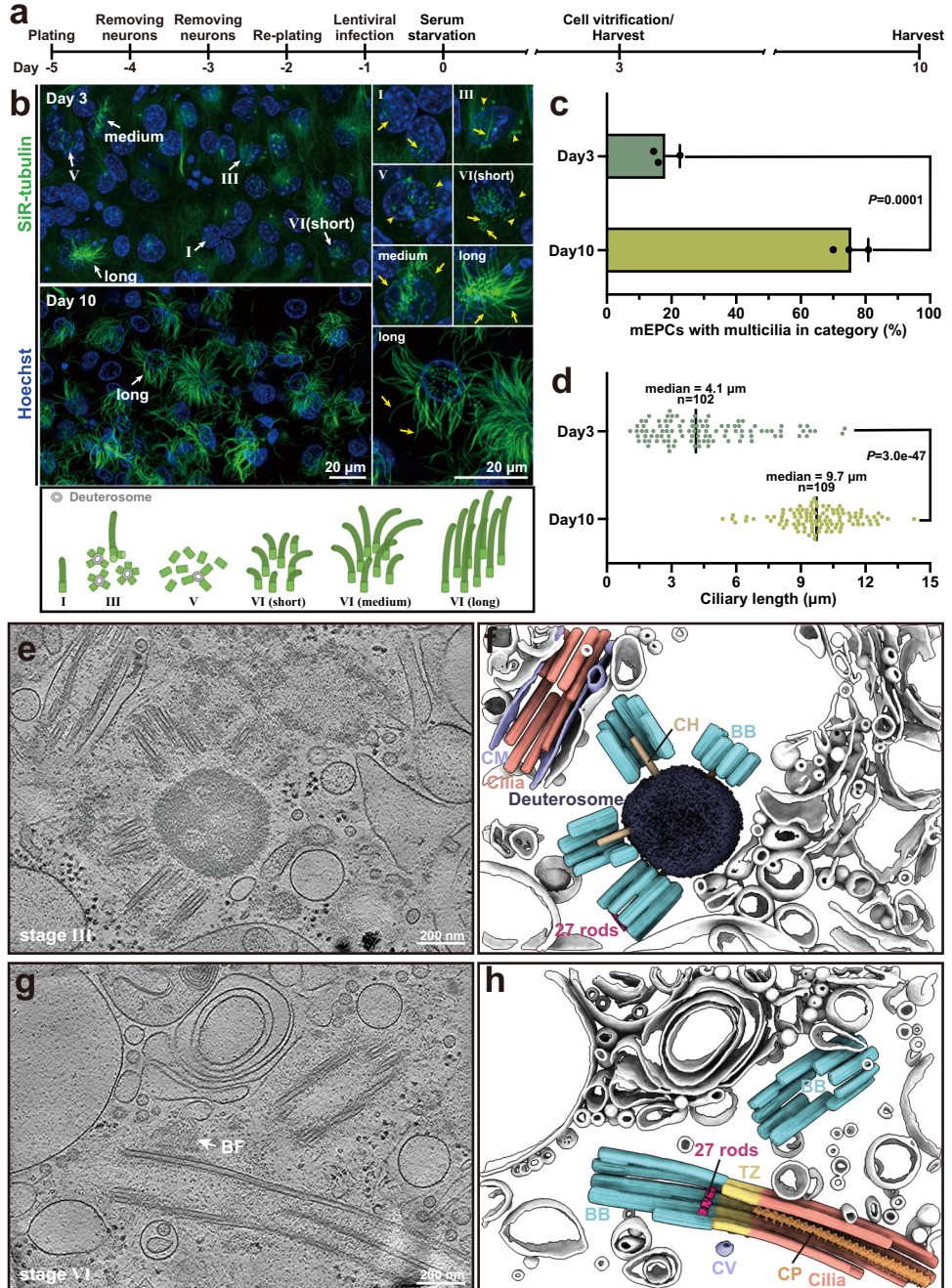

**Fig. 1 | Deuterosome-dependent basal body biogenesis and ciliogenesis in mEPCs. a** Schematic representation of the experimental timeline. Cells from mouse telencephalon tissues were cultured and induced to differentiate into mEPCs through serum starvation. **b** Representative day-3 and day-10 mEPCs were stained for microtubules (SiR-tubulin, green) and nuclei (Hoechst 33342, blue) (top). White arrows highlight magnified regions (2×) showcasing cilia (yellow arrows) and nascent BBs (yellow arrowheads). Stages of mEPCs in massive BB biogenesis are indicated with Roman numerals and illustrated with the cartoon model (bottom). Stage-I cells contain solely parental centrioles and 1–2 primary cilia; stage-III cells feature multiple procentrioles around deuterosomes and parental centrioles; stage-V cells are releasing nascent BBs from deuterosomes and parental centrioles; and stage-VI cells contain mature BBs and also short multicilia[32,33]. Following the ciliogenesis, multicilia further elongate. The short,

medium, and long denote the relative multicilia length of 0–5.37 μm, 5.37–9.89 μm, and 9.89–14.25 μm, respectively. **c** Abundance of multiciliated cells. Quantification results from three independent experiments are presented as mean ± SD. At least 84 cells were scored for each experiment and condition. *P-value* was derived from two-sided student's *t*-test. **d** Lengths of multicilia. The average length of three longest cilia per multiciliated cell was used to represent the length of the multicilia. At least 102 mEPCs in stages II–IV from two independent experiments were scored. *P-value* was derived from two-sided student's *t*-test. **e**, **g** 2.7-nm-thick tomographic slices represent BB biogenesis at stage III (**e**) and VI (**g**). The white arrow indicates basal foot (**g**). **f**, **h** 3D rendering of tomograms corresponding to the regions of (**e**) and (**g**). BF basal foot, CM ciliary membrane, CV ciliary vesicle, CH central hub, BB basal body, TZ transition zone, CP central pair. Source data are provided as a Source Data file.

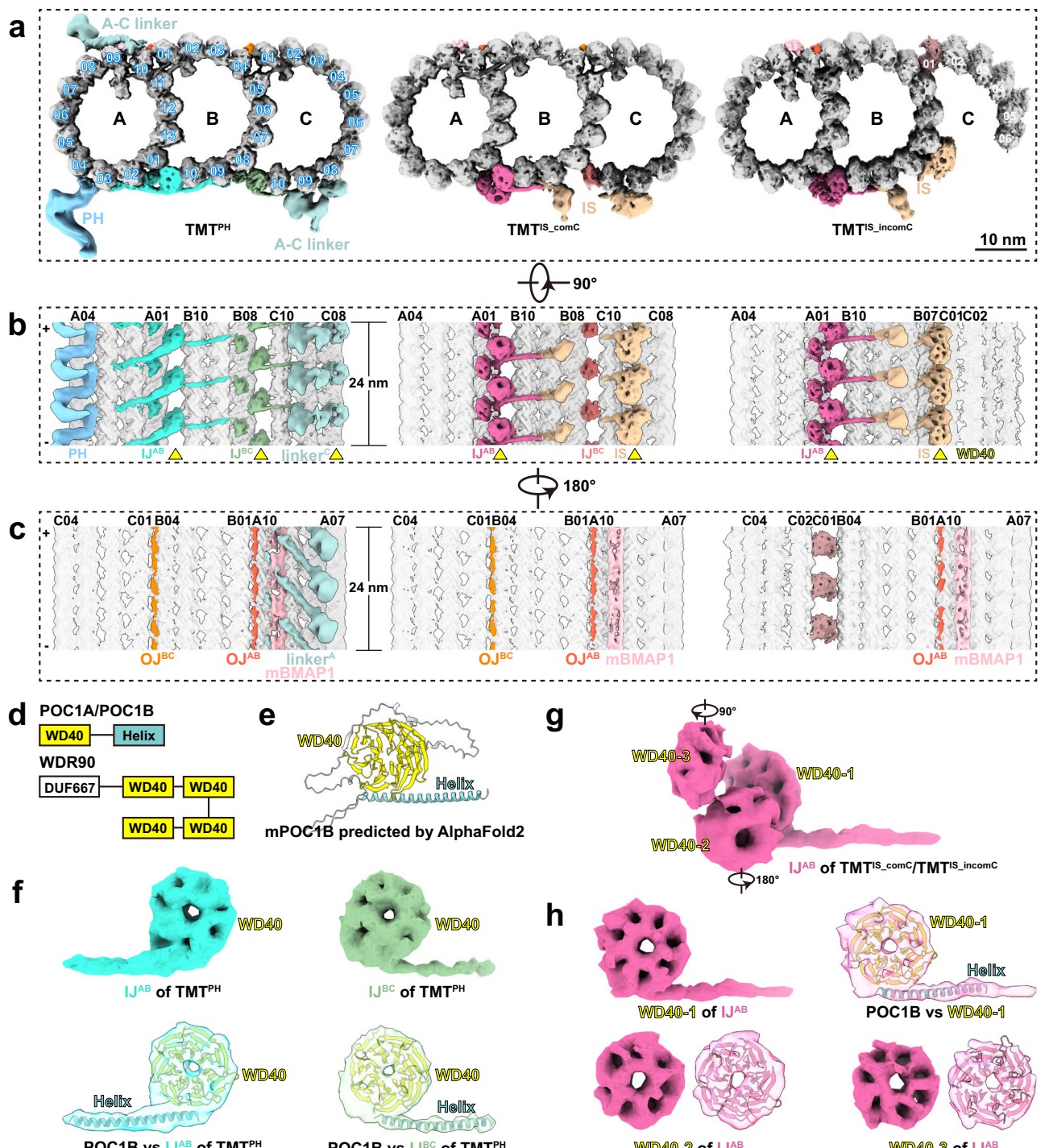

**Fig. 2 | The longitudinal variations and distinct inner junctions within basal bodies. a–c** Cross-sectional views from the plus end (**a**), luminal side (**b**) and extraluminal side (**c**) of cryo-EM density maps of TMT^PH (left), TMT^IS_comC (middle) and TMT^IS_incomC (right) within basal bodies. Protofilaments are numbered. The minus (−) and plus (+) ends of TMT are indicated. Yellow triangles annotate the features of WD40 domains. **d** Secondary structure diagrams of mouse POC1B and WDR90. **e** The cartoon model of AlphaFold2-predicted mouse POC1B, colored according to the secondary structure in (**d**). **f** Enlarged views of the segmented density maps (top) and rigid-body fitting of AlphaFold2-predicted mouse POC1B into the corresponding densities (bottom) of IJ^AB and IJ^BC within TMT^PH. The density maps are shown as transparent surfaces, with residues outside the densities omitted for clarity (bottom). **g, h** The density map of IJ^AB in the inner scaffold region (TMT^IS_comC/TMT^IS_incomC) (**g**) and the magnified views of three WD40 domains (**h**). The rotation directions and angles in (**g**) annotate the views of WD40-2 and WD40-3 in (**h**). The AlphaFold2-predicted mouse POC1B was fitted into the density of WD40-1, while only WD40 domains were fitted into the densities of WD40-2 and WD40-3 (**h**). PH pinhead, IS inner scaffold, IJ inner junction, OJ outer junction.

the critical junction between TMT^PH and the cartwheel (Fig. 2a, b)[20,43]. CEP135, predicted to be a major component of the pinhead[46,47], consistently persisted in mTECs and mEPCs during ciliogenesis[39], in agreement with the persistent presence of the pinhead in BBs,

regardless of the existence of the cartwheel (Fig. 2a, b and Supplementary Fig. 2a, b).

The inner scaffold, lagging above the region of the cartwheel and spanning from the central (TMT^IS_comC) to distal (TMT^IS_incomC) regions of

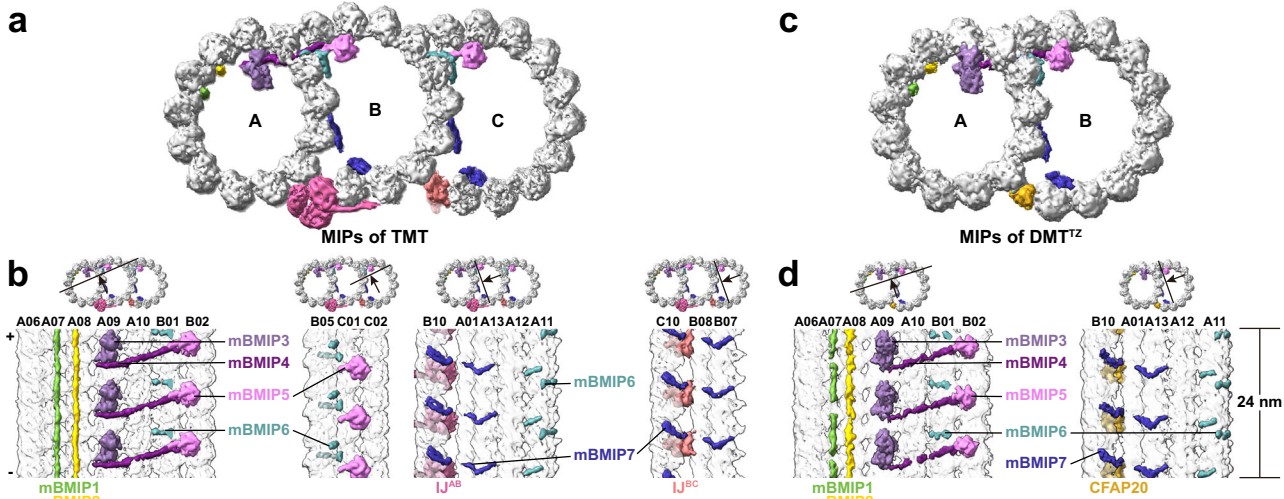

**Fig. 3 | Conserved 8-nm repeat MIPs between TMTs within the basal body and DMTs within the transition zone. a**, **c** Density maps of TMT within BBs (**a**) and DMT within the TZ (**c**) show the conserved MIPs. The structure of TMT$^{IS\_comC}$ was used to represent the overall structure of BB, given the consistent MIPs along BBs. Densities outside tubules of TMT and DMT are omitted for clarity. **b**, **d** The cross sections (top) show the TMT (**b**) and DMT$^{TZ}$ (**d**) map. The longitudinal sections (bottom) show the 8-nm-repeat MIPs. Protofilaments are numbered. Within BBs (**b**), the C tubule-binding MIPs were absent in TMT$^{IS\_incomC}$. Polarities of TMTs or DMTs are indicated with the plus (+) and minus (−) symbols. Although with different periodicities, mBMIP1 resembles CFAP53 in DMTs; mBMIP2, MNS1; mBMIP3, NME7; mBMIP4, CFAP141; mBMIP7, ENKUR.MIP microtubule-inner protein, IJ inner junction.

BBs, attaches to the luminal side of BBs and plays a key role in maintaining BB structural integrity (Fig. 2a, b and Supplementary Fig. 2a). Along the proximal-to-distal axis of BBs, the loss of C-tubule protofilaments initiates at C10 and proceeds to C01 (Supplementary Fig. 2c). The averaged structure of TMT$^{IS\_incomC}$ is featured by the incomplete C tubule with six protofilaments, as evidenced by its prolonged existence in tomograms, and C01 is composed of an unknown protein at an 8-nm periodicity (Fig. 2a, c and Supplementary Fig. 2c). The incompleteness of the C tubule in this region renders its inner scaffold-binding modules to adopt an inward orientation, enabling them to interact with B07-B08 on the exterior of the B tubule (Supplementary Fig. 2d). Besides, both TMT$^{IS\_comC}$ and TMT$^{IS\_incomC}$ are decorated by two striations that laterally stretch across the protofilaments outside A and C tubules at 8-nm intervals (Supplementary Fig. 2d). The striation that spans C04-C07 in TMT$^{IS\_comC}$ slides to C02-C05 in TMT$^{IS\_incomC}$, while the striation outside A tubule remains unchanged. These structural disparities between TMT$^{IS\_comC}$ and TMT$^{IS\_incomC}$ likely correlate with the gradual loss of C tubule in distal BBs, although the specific process remains elusive.

Together, we subdivided the BB into three regions based on the structural variations of TMTs from proximal to distal ends, which were featured by the pinhead and inner scaffold with complete or incomplete C tubule, respectively. These distinct structures reveal that the 8-nm-repeat MAPs collectively construct the ninefold symmetrical configuration of BBs and their remodeling contributes to the progressive transition from proximal to distal BBs, as well as the reduction of C tubule.

## Multiple WD40 domain-containing proteins within BBs

Structural analysis disclosed that various doughnut-shaped densities widely spread at the inner junctions, A-C linker and inner scaffold of BBs (Fig. 2a, b), aligning with the configuration of the WD40 domain, a seven-bladed β-propeller. The application of expansion microscopy has identified several proteins in the central region of centriole, including POC1B, WDR90, FAM161A and POC5[25,48], among which POC1B and WDR90 possess one and four WD40 domains, respectively (Fig. 2d, e). Besides, POC1B has been promoted to be an inner junction protein throughout BBs in *Tetrahymena*[49,50].

Within TMT$^{PH}$, both inner junctions of A-B tubules (referred to as IJ$^{AB}$) and B-C tubules (referred to as IJ$^{BC}$) are composed of a core WD40 domain and connecting α-helices (Fig. 2b, f). Rigid body fitting of the AlphaFold2[51,52] predicted mouse POC1B fitted well in the densities of IJ$^{AB}$ and IJ$^{BC}$ of TMT$^{PH}$ (Fig. 2f). The lateral IJ$^{AB}$ and IJ$^{BC}$, spanning from A03 to B09, connect pinhead and A-C linker and potentially stabilize the microtubule lattice (Fig. 2b). Furthermore, an additional WD40 domain is identified in the linker$^C$ of A-C linker (Fig. 2b and Supplementary Fig. 2e), which is potentially contributed by WDR67, a component of the A-C linker identified through ultrastructure expansion microscopy (U-ExM)[41]. Given that previous cryo-ET investigations in *Chlamydomonas* have identified the presence of a WD40 domain in the linker$^A$ of A-C linker[21], it is proposed that the WD40 domain may enhance the rigidity of the proximal cartwheel region within BBs.

While the core WD40 domain (WD40-1) of IJ$^{AB}$ within TMT$^{IS\_comC}$ and TMT$^{IS\_incomC}$ remained minimally changed compared to that within TMT$^{PH}$, two extra WD40 domains (WD40-2 and WD40-3) were revealed in the IJ$^{AB}$ of the inner scaffold region (Fig. 2g, h). WDR90, with four WD40 domains (Fig. 2d) and confined localization in the region[48,50], potentially contributes to the two WD40 domains. On the contrary, the density of IJ$^{BC}$ becomes weaker in TMT$^{IS\_comC}$ and completely absent in TMT$^{IS\_incomC}$ (Fig. 2b), possibly due to the structural heterogeneity of C tubule in this region. Besides, two unassigned densities, composed of two WD40 domains, were observed at C09-C10 of TMT$^{IS\_comC}$ and B07-B08 of TMT$^{IS\_incomC}$, respectively (Fig. 2b and Supplementary Fig. 2d, f, g), constituting a portion of the inner scaffold.

Collectively, although the protein identities of different WD40 domains are challenging to identify, their intimate interconnections constitute the foundational architecture of inner junctions, A−C linker and inner scaffold, which noticeably enhance the integrity and stability of BBs.

## A unique MIP repertoire of BB

Different from the densities outside microtubules, the 8-nm-repeat MIPs remain consistent longitudinally along BBs except for the empty C tubule of TMT$^{IS\_incomC}$ (Figs. 2a and 3a, b). For clarification, α/β-tubulin dimers of mouse DMTs[12] were rigid body-fitted to the TMT backbone and extra densities were given the names mBMIP1-7. TMT exhibits

nearly hollow tubules with few MIPs, whose arrangement is more likely a simplified version of DMT, given the resemblance of most MIPs (mBMIP1-4 and mBMIP7) to those in DMT[12,14,16] (Fig. 3a, b and Supplementary Fig. 2h). For instance, the typical A09-binding MIP, namely NME7 in DMT, exhibits a 48-nm periodicity with 16- and 32-nm intervals[12,14,16,17], which, nevertheless, is streamlined to a pure 8-nm-repeat mBMIP3 within BBs (Fig. 3a, b and Supplementary Fig. 2h). Besides, mBMIP5 occupies a distinct tubulin-binding location at B02 and C02 of TMT (Fig. 3a, b), which has been observed in the structures of centrioles from CHO cells and cilia from bovine tracheae[22,26]. We postulate CEP41 as a potential candidate of mBMIP5, which will be discussed later.

## Plasticity of ciliogenesis pathways in multiciliated cells

After the construction of BBs, ciliogenesis occurs by forming the TZ, followed by the elongation of axonemal DMTs[30]. Interestingly, while multiciliated cells are thought to favor the extracellular pathway during ciliogenesis[8,9], our observations in mEPCs revealed a more nuanced picture. As revealed in some of our tomograms, upon dissociation from deuterosomes, BBs can directly dock at the apical plasma membrane, allowing the axoneme to extend outwards (Supplementary Fig. 3a, b). This extracellular pathway provides a direct way for cilia to anchor on the cell surface. On the other hand, bare axonemal microtubules with associated BBs and surrounding ciliary vesicles were also observed in the cytoplasm, suggesting the existence of the intracellular pathway (Fig. 1g, h and Supplementary Fig. 3c). The fusion of ciliary vesicles leads to the construction of submerged cilia capped by a ciliary sheath (Supplementary Fig. 3d). In our analysis of 261 tomograms, 38 tomograms were categorized as the extracellular pathway, characterized by cilia docked to the apical membrane. Additionally, 26 tomograms were classified under the intracellular pathway, distinguished by cilia either surrounded by vesicles or capped by the ciliary sheath within the cytoplasm. These findings highlight the remarkable plasticity and adaptability of ciliogenesis pathways in multiciliated cells.

In addition, despite that the flexibility of Y-links makes them challenging to observe in our tomograms, the ciliary necklace, strands of membrane particles that appear to anchor the ends of Y-links[10,53], was revealed at the TZ region (Supplementary Fig. 3a–d). Similar particles were observed sporadically on the BB-docking plasma membrane (Supplementary Fig. 3a), ciliary membrane beyond the TZ (Supplementary Fig. 3b), and ciliary vesicles (Supplementary Fig. 3c), implying possible sources of the particles and assembly pathways of the necklace during ciliogenesis.

## Microtubule-binding proteins in TZ region feature transitions from TMT to DMT

Although our tomograms did not resolve Y-links, STA enabled structural determination of the DMT in the TZ region (referred to as DMT$^{TZ}$) to 12.6 Å, the highest resolution to date[26] (Supplementary Fig. 1i, k). In the structure of DMT$^{TZ}$, densities binding to A06-A08 outside the A tubule resemble the A-B linker, and the striation laterally spanning outside the B tubule is likely transformed from the C-tubule striations of TMT$^{IS\_comC}$ and TMT$^{IS\_incomC}$ (Supplementary Fig. 3e). Moreover, the periodicities and identities of DMT$^{TZ}$ MIPs remain the same as those of TMT except for the inner junction (Fig. 3). The inner junction of DMT$^{TZ}$ consists solely of CFAP20 with an 8-nm periodicity, lacking both the WD40 domain-containing proteins within BBs and the PACRG in ciliary axonemes[12,14,16] (Fig. 3c, d). In addition, a coiled-coil density mBMAP2 fills the gap between A02 and A03 of DMT$^{TZ}$ (Supplementary Fig. 3e), resembling CCDC39 and CCDC40 in ciliary DMT[13]. Although the 8-nm periodicity of DMT$^{TZ}$ prevents the identification of mBMAP2, the above protein transitions of inner junctions and MAPs suggest that DMT$^{TZ}$ appears to be an intermediate state between the TMT and canonical DMT, and

further paint a picture of protein remodeling from TMT to DMT within the TZ region.

## The TMT-DMT interface is bordered by the 27 rods and associated structures

Could there be structures that spatially define the boundary between the BB and the axoneme, or, in other words, TMT and DMT? An annular structure composed of 27 evenly-distributed rod-like structures (27 rods) has been shown previously to reside in the central lumen of centriolar distal end in HeLa cells[54,55]. We observed that a similar structure, with ~100 nm in diameter and 35 nm in height, consistently sited at the distal end of BBs in mEPCs (Fig. 4a–c). To further validate its biogenesis timing, we examined different BBs under assembly, i.e., containing the central hub of the cartwheel, and observed the 27 rods in all such BBs except for one containing only TMT$^{PH}$, with the minimal distance to BB's proximal end being about 160 nm (Fig. 4d and Supplementary Fig. 2b). Furthermore, the 27 rods persisted in BBs docked at the apical membrane or with axonemes (Fig. 4e, f and Supplementary Fig. 2b). As the 27 rods were observed within the regions of TMT$^{IS\_comC}$ or TMT$^{IS\_incomC}$ but not TMT$^{PH}$ (Supplementary Fig. 2b), we reason that they are assembled after TMT elongations pass the cartwheel region and reside persistently on the growing distal end during the BB biogenesis.

We then performed STA analysis for the rods and found that they are close to TMTs via a T-shaped density on the A tubule (Supplementary Fig. 4a–c), consistent with observations in HeLa cells[55] (Supplementary Fig. 4d). While the T-shaped density appears to contact with three adjacent rods, there is a 6-nm gap between them, probably filled with unidentified flexible proteins (Supplementary Fig. 4e). Moreover, the tail of the T-shaped density extends from A05 outside the A tubule to anchor at A09 and forms an 8-nm periodicity at the extreme end of TMTs (Supplementary Fig. 4f). We thus infer that it is these structures that continuously track the plus end of TMTs to achieve the persistent distal end positioning of the 27 rods during BB biogenesis (Fig. 4 and Supplementary Fig. 2b). The perfect alignment of the 27 rods and related structures with the TMT-to-DMT transition (Fig. 4f and Supplementary Figs. 2b, 4f) indicates that they altogether form a definitive TMT-DMT boundary, or BB-to-TZ demarcation point in cilia: during ciliogenesis, TZ region initiates from the plane defined by the 27 rods.

## Structural differences between nascent and mature DMTs

Our ex vivo system captured ciliogenesis at an early stage, as day-3 mEPCs possessed fewer and shorter cilia compared to day-10 cells (Fig. 1b–d). Initial reconstruction of the 96-nm-repeat structure of DMT (referred to as DMT$^{96nm}$) unveils a partial assembly of axonemal complexes including radial spokes (RS), nexin-dynein regulatory complexes (N-DRC), inner dynein arms (IDA) and outer dynein arms (ODA), along with multiple holes in the inner junction (Supplementary Fig. 5a).

The 48-nm-periodic structure of DMT (referred to as DMT$^{48nm}$) further reveals a nascent form compared with the structure in mature cilia from day-10 mEPCs (EMD-37111) published recently[19] (Fig. 5a–c). Specifically, the inner junction protein PACRG exhibits a distinct 16-nm periodicity, different from its 8-nm form in mature cilia (Fig. 5b and Supplementary Fig. 5c)[12–14,19]. Besides, DMT$^{48nm}$ from day-3 mEPCs has two more MIPs: A09-binding MIP with a 48-nm periodicity and B02-binding MIP with an 8-nm repeat (Fig. 5b and Supplementary Fig. 5c). The former may be additional copies of NME7, similar to mBMIP3, and is thus referred to as mBMIP3L for mBMIP3-like (Fig. 5b and Supplementary Fig. 5b, c). The latter mirrors mBMIP5 at B02 and C02 in the BB and TZ but with weaker density (Figs. 3 and 5b). Focused classification on mBMIP5 of DMT$^{48nm}$ revealed two distinct maps: one with mBMIP5 and another lacking it (Fig. 5c). Strikingly, the absence of mBMIP5 density correlates with a lack of mBMIP3L and increased ODA density, suggesting a more advanced stage closer to mature DMT (Fig. 5c). The

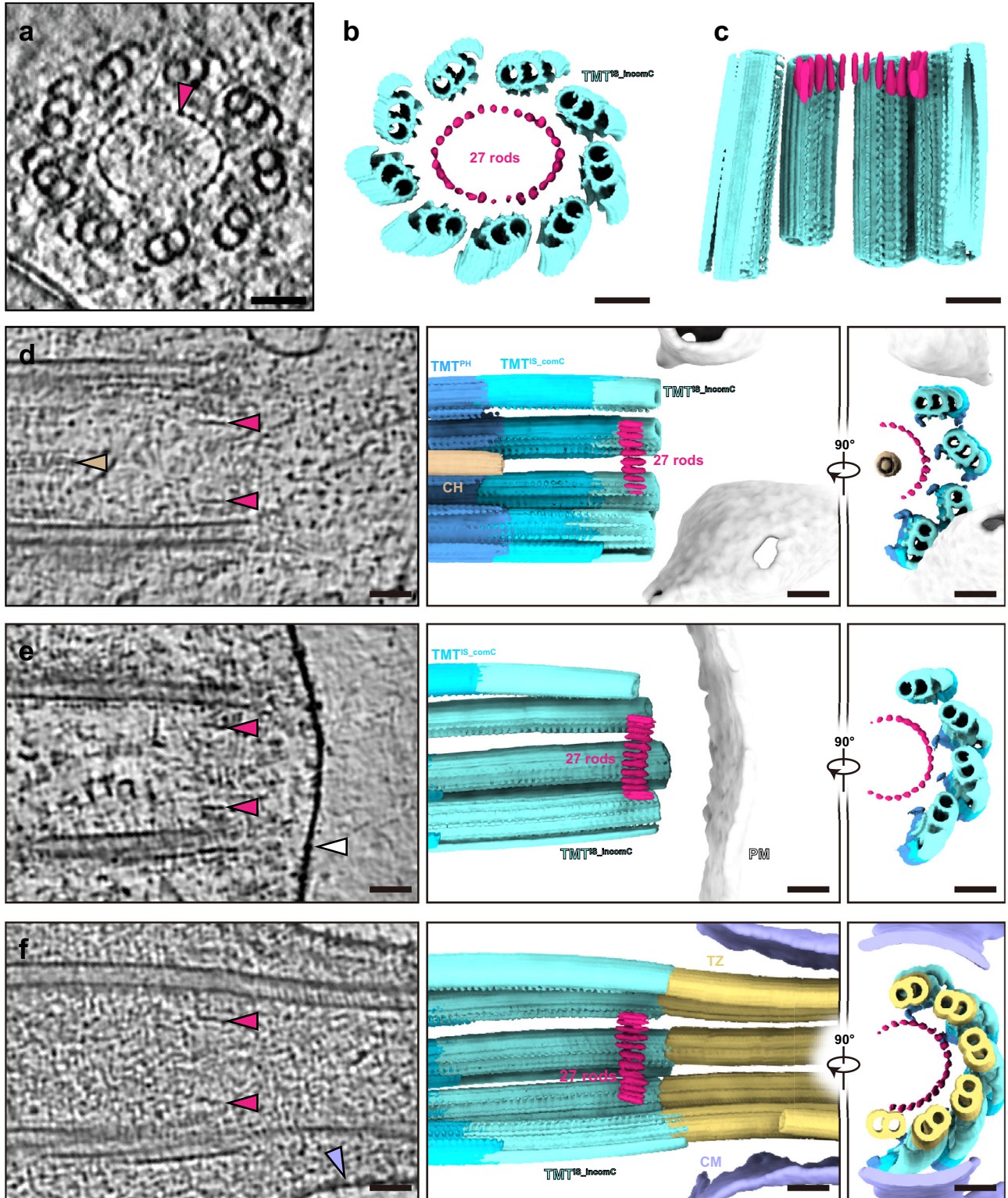

**Fig. 4 | The 27 rods reside at the distal end of basal bodies. a** The 20-nm-thick tomographic slice displays the cross-section view of 27 rods (red arrowhead) within the central lumen of basal body. **b, c** 3D renderings showcase the cross-section view and clip view of structures shown in (**a**). **d–f** 20-nm-thick tomographic slices (left) and 3D renderings (middle and right) display that the structure of 27 rods residing at the distal end of basal bodies in three progressive states: basal body under assembly (**d**), basal body docking at the plasma membrane (**e**) and basal body templating a cilium (**f**). 27 rods (red), central hub (tan), plasma membrane (white) and ciliary membrane (purple) were indicated by arrowheads. Scale bar, 50 nm. CH central hub, PM plasma membrane, CM ciliary membrane, TZ transition zone.

incomplete assembly of PACRG and axonemal complexes in nascent cilia, as well as the gradual loss of mBMIP5 in DMT$^{TZ}$, nascent cilia and mature cilia, further underlines the potential protein rearrangements and the progressive maturation of cilia as axoneme extends[56].

To further validate this hypothesis, we directly assessed the ciliary motility in mEPCs using high-speed fluorescent microscopy. Surprisingly, multicilia beat in a back-and-forth manner, the characteristic beat pattern, even when they are short, revealing the inherited

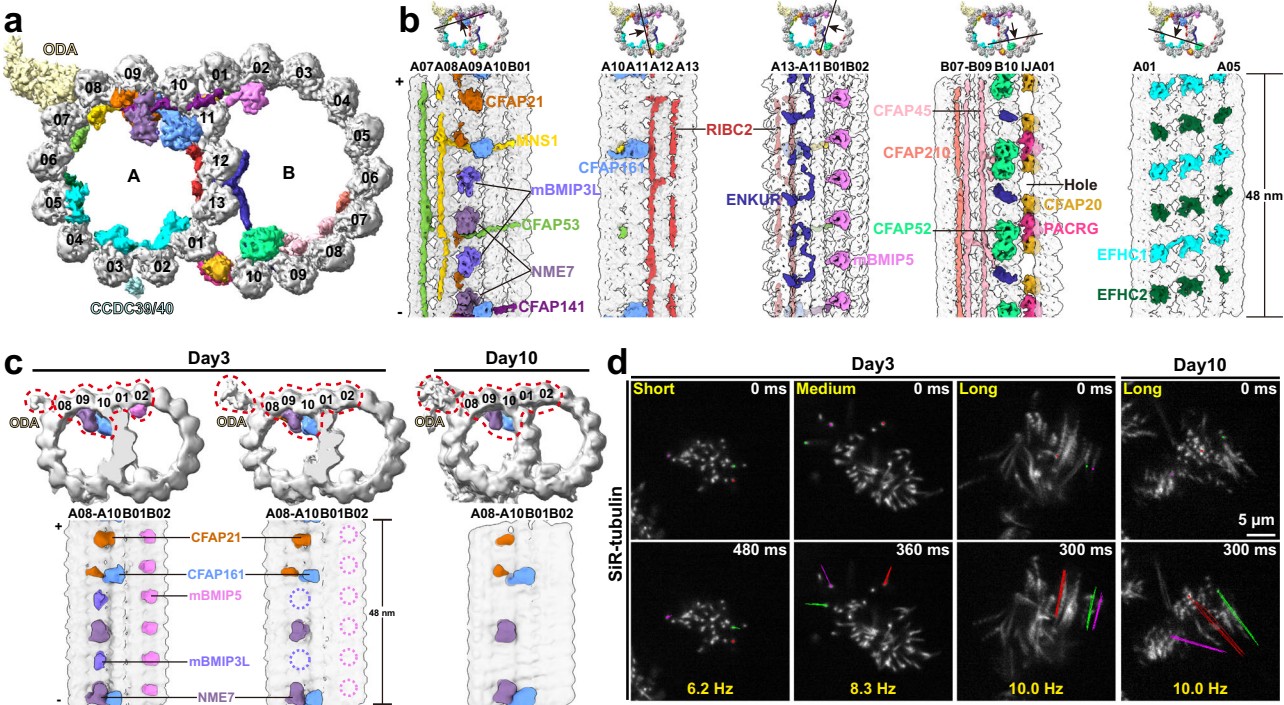

**Fig. 5 | Difference between the nascent and mature microtubule doublets. a** The cross-section view of the DMT$^{48nm}$ density map. Protofilaments are numbered accordingly. The MIPs and ODA are segmented and colored as indicated in (**b**). **b** The cross sections (top) show the DMT$^{48nm}$ density map, indicating the location of longitudinal sections (bottom). The longitudinal sections (bottom) with the MIPs segmented and colored as indicated. The site of absent PACRG in nascent cilia is indicated. The minus (-) and plus (+) ends of DMT are indicated. IJ inner junction. **c** Focused classification on mBMIP5 of DMT from day-3 mEPCs generated two maps

(left and middle). The dashed circles (middle) indicate the shedding of corresponding MIPs (mBMIP5 and mBMIP3L) of the same color. The DMT without mBMIP5 is similar with that from day-10 mEPCs (EMD-37111). **d** Motilities of representative cilia. Multicilia in living mEPCs were stained with SiR-tubulin, followed by high-speed imaging at 60-ms (day-3 cilia) or 50-ms (day-10 cilia) intervals. At least 50 multiciliated cells were captured each experiment. Beat tracks of three traceable cilia in three beat cycles are shown in color for each cluster of representative multicilia.

robustness of the beat machinery (Fig. 5d and Supplementary Movies 2–5). The ciliary beat frequencies appear to increase upon cilia elongation (Fig. 5d). Furthermore, short or medium cilia beat in different directions, whereas long cilia tend to beat coordinately.

These results provide progressive structural landscapes for cilia that are undergoing construction and modulation, and suggest that the initial recruitment and subsequent removal of mBMIP3L and mBMIP5 respectively impact DMT assembly or construction of the robust beat machinery. Consistently, depletion of NME7, a nucleoside diphosphate kinase fitted into the density maps of both mBMIP3 and mBMIP3L (Supplementary Figs. 2h and 5b), in rats has recently been shown to cause ciliopathies, including severe hydrocephalus, reversed internal organ arrangement, and sterility[57].

**The density of mBMIP5 is assigned to CEP41, a regulator of motile cilia formation**
To verify the role of mBMIP5, we refined the corresponding region of TMTs to achieve a resolution of 8.1 Å, allowing clear visualization of α-helices (Fig. 6a). We then systematically docked AlphaFold2 models of the entire mouse proteome into the mBMIP5 density map (Supplementary Fig. 6a). CEP41 emerged as the best match based on cross-correlation values and manual inspection (Supplementary Fig. 6a). Notably, the rigid core of CEP41 with a reliable predicted structure fitted well with the mBMIP5 density, including a rhodanese-like domain (RHOD) and several helices (Fig. 6b, c and Supplementary Movie 6), while the density of the flexible loops was not resolved in our reconstruction (Supplementary Fig. 6a).

CEP41 is a BB- and primary cilium-localized protein mutated in patients with Joubert syndrome[58,59]. Exogenously-expressed CEP41

exhibits an annular distribution along the centriole axis in U2OS cells[60]. Analysis of our previous microarray results of mTECs[61] suggested that *CEP41* is upregulated following the differentiation into multiciliated cells (Supplementary Fig. 6b). Its mRNA levels peaked at around day 3, when mTECs were actively undergoing massive BB biogenesis and multiciliogenesis[32], and then gradually declined (Supplementary Fig. 6b). Immunoblotting indicated high-levels of CEP41 in mEPCs and mouse testis and relatively low-levels in IMCD3 cells and mouse brain (Supplementary Fig. 6c). Therefore, the varied expression of CEP41 suggests its potential correlation with motile cilia formation.

Immunostaining indicated localizations of CEP41 in both BBs and axonemes of multiciliated mEPCs (Fig. 6d). Furthermore, its axonemal localizations could be classified into three types. In mEPCs with short or medium cilia (Fig. 1b, d), CEP41 usually fully decorated the axonemes (Fig. 6d). In the long cilia of day-3 mEPCs (Fig. 1b, d), CEP41 preferentially localized to the tip region (Fig. 6d). In the long cilia of day-10 mEPCs, however, CEP41 was typically enriched in a narrow subdistal region (Fig. 6d). We also noted that even in day-10 mEPCs with long cilia, there were some short axonemes fully decorated with CEP41 (Fig. 6d). We quantified the lengths of these axonemes and found that they fell into categories of short or medium axonemes ($3.0 \pm 0.9\,\mu m$) as compared with those with subdistal CEP41 ($7.9 \pm 1.2\,\mu m$) (Figs. 1d and 6e), indicating that these axonemes belong to newly-assembling cilia due to the persistent ciliogenesis in multiciliated mEPCs[62]. Taken together, we conclude that CEP41 displays persistent BB localization and its axonemal distributions alter dynamically from the initially full axoneme to the axonemal tip region and eventually a subdistal region following the maturation of motile cilia (Fig. 6f). These observations are consistent with the behaviors of

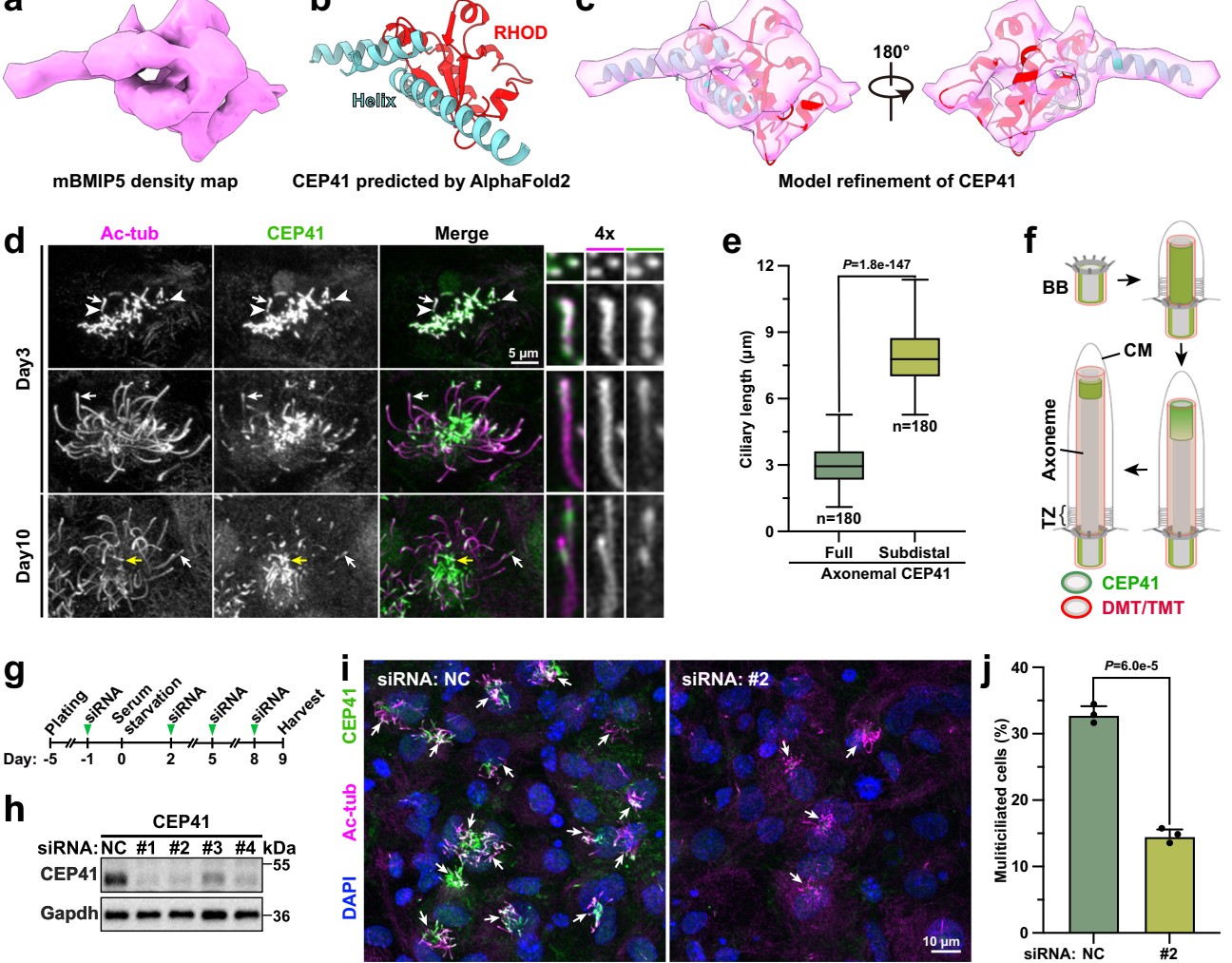

**Fig. 6 | CEP41 is required for multiciliogenesis. a** The mBMIP5 density map. **b** Cartoon model of AlphaFold2-predicted mouse CEP41 with the rhodanese-like domain (RHOD, red) and α-helices (cyan). Disordered loops were omitted. **c** Refined model of mouse CEP41 fitted into the mBMIP5 density. Residues outside the density were omitted. **d** CEP41 relocated from full axonemes to a subdistal region during motile cilia elongation. Day-3 and day-10 mEPCs were immunostained with CEP41 and acetylated tubulin (Ac-tub) to visualize axonemes and BBs. CEP41 is mainly distributed along axonemes (top), at the tip region (middle), or in a subdistal region (bottom). Zoom-in views of representative BBs (white arrowheads) or axonemes (white arrows) were shown. Yellow arrows denote the axonemes fully decorated with CEP41 in day-10 mEPCs. **e** Axonemal length distributions with full or subdistal CEP41 localization. 180 axonemes of multiciliated day-10 mEPCs from three independent experiments were used for quantification. The boxes span the

interquartile range (IQR, 25th to 75th percentiles), with a central line marking the median. Whiskers extend to maximum and minimum data points within 1.5×IQR from the quartiles. No outliers beyond the whiskers are observed. *P-value* was derived from two-sided student's *t*-test. **f** Illustrations for CEP41 localizations in motile cilia formation. BB basal body, TZ transition zone, CM ciliary membrane. **g, h** Experimental scheme of CEP41 knockdown (**g**) and expression patterns (**h**) in mEPCs. Four biologically independent experiments were performed. NC negative control. **i, j** CEP41 depletion repressed multiciliogenesis. mEPCs were immunostained for CEP41 and Ac-tub and counterstained with DAPI to visualize the nucleus (**i**). Incidences of multiciliated cells (white arrows) from three independent experiments are presented as mean ± SD (**j**). At least 188 cells were scored in each experiment and condition. *P-value* was derived from two-sided student's *t*-test. Source data are provided as a Source Data file.

mBMIP5 (Figs. 3 and 5), which is initially recruited to nascent DMT and subsequently displaced during maturation, further supporting the assignment of mBMIP5 to CEP41.

To assess the functions of CEP41 in motile cilia, we performed RNAi to knock down the protein in mEPCs (Fig. 6g–i) using previously reported siRNAs against *CEP41* mRNA[58]. Examinations on mEPCs treated with siRNA #2 or #4 revealed striking reductions in multiciliated cells at day 9 (Fig. 6i and Supplementary Fig. 6d). Quantifications indicated a 2.2-fold reduction on average in siRNA #2-treated mEPCs as compared to mock-treated controls (Fig. 6j). Furthermore, in multiciliated mEPCs lacking CEP41, ciliary lengths were generally incomparable to those of control mEPCs (Fig. 6i and Supplementary Fig. 6d). Although the RNAi did not abolish the multicilia formation possibly due to partial knockdown, as evidenced by the presence of

residual CEP41 (Fig. 6h and Supplementary Fig. 6c), the reduction in multiciliated cells and cilia length indicated that CEP41 is required for normal multiciliogenesis.

## Discussion

Cilia play important roles in signal transduction and cell motility, with their dysfunctions underlying many human disorders[5]. Ciliogenesis is a complicated process involving hundreds of proteins under sophisticated regulation[8]. To date, while the structures of DMT from different species or tissues have been extensively resolved[12–17], the studies were primarily attributed to the ex vivo purification of mature DMT. Limited structural studies on the integral BB biogenesis as well as ciliogenesis, especially in metazoa, restrict our understanding of the underlying assembly mechanism. This study fills the gap by establishing a practical

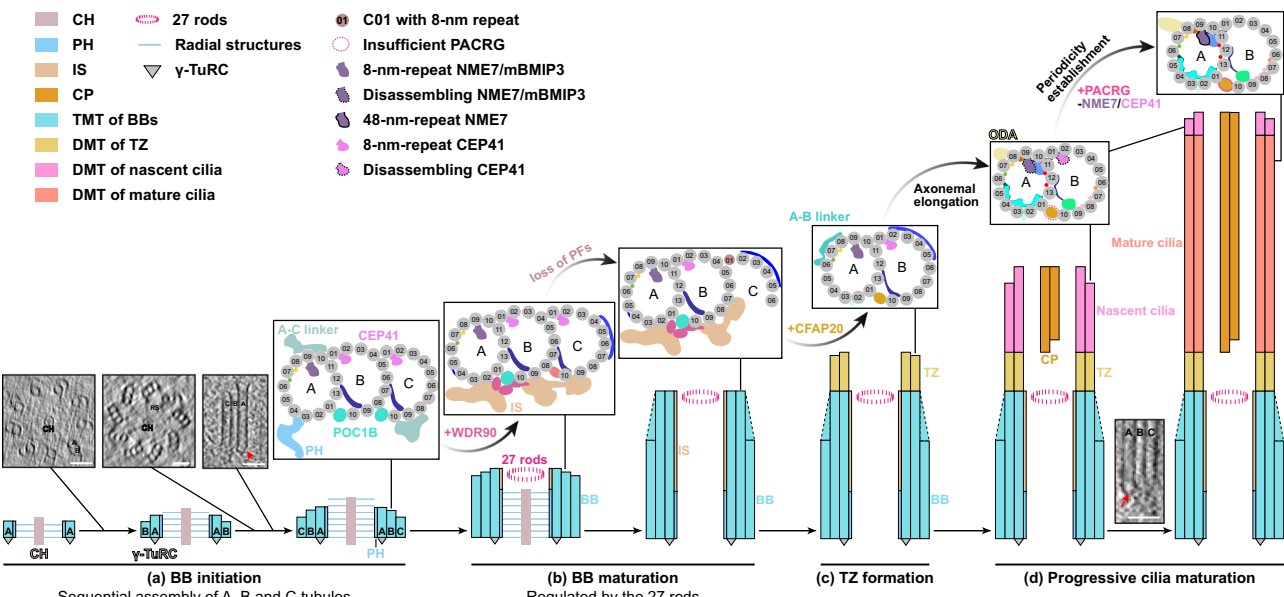

**Fig. 7 | Model of the regulated basal body biogenesis and ciliogenesis.** The step-wise multiciliogenesis in mEPCs encompasses four phases: BB initiation, BB maturation, TZ formation and progressive cilia maturation: **a** During BB initiation, the cartwheel emerges first, followed by the sequential assembly of A, B and C tubules, as indicated by the 54- and 27-nm-thick tomographic slices (left and middle). The γ-TuRC (red arrow) anchors at the minus end of A tubule from an assembling BB, as illustrated by the 16-nm-thick tomographic slice (right). The cartwheel emanates nine radial structures binding to the pinhead of TMTs while neighboring TMTs are interconnected by the A–C linker; **b** After TMTs extend beyond the cartwheel region, BB assembly is capped by the distal luminal 27 rods. The helical inner scaffold attaches to the luminal side and integrates the BB into a unity, whereas the inner and outer junction proteins contribute to the longitudinal variations observed along BBs. Following BB elongation, nascent C tubule gradually lacks protofilaments (dotted lines); **c** Mature BBs template ciliogenesis by TZ formation. The annual structure of 27 rods serves as a definitive boundary between TMTs and DMTs. The DMT in TZ is featured by the substitution of inner junction proteins with 8-nm-repeat CFAP20; **d** As the axoneme elongates, microtubule-binding proteins, especially MIPs, undergo remodeling to establish the canonical 48-nm periodicity. This is exemplified by the gradual incorporation of PACRG and the shedding of A09-binding NME7/mBMIP3 and B02-binding CEP41. The tip of mature cilia undergoes constant turnover, evidenced by the CEP41 distribution. The tomographic slice from Fig. 1g highlights the persistent presence of γ-TuRC (red arrow) during ciliogenesis. Scale bar, 50 nm. CH central hub, RS radial structures, PH pinhead, IS inner scaffold, CP central pair, BB basal body, TZ transition zone, γ-TuRC γ-tubulin ring complex.

pipeline in mEPCs using cryo-CLEM, cryo-FIB, and cryo-ET, which allows us to capture the biogenesis of BBs and cilia at various stages in situ. We show that both intracellular and extracellular pathways are adopted for efficient multiciliogenesis (Supplementary Fig. 3a–d), revising current views[8,9]. More importantly, the unprecedented details are distilled into an elaborate model recapitulating the step-wise assembly process encompassing BB initiation, BB maturation, TZ formation and progressive cilia maturation (Fig. 7). Notably, there are several similarities between BB initiation and procentriole formation: the protruding cartwheel forms prior to TMTs; the A, B and C tubules are assembled sequentially; and the minus end of A tubule is capped and potentially nucleated by the γ-tubulin ring complex (γ-TuRC) (Fig. 7). Unlike the missing γ-TuRC in human mature centrioles[24,41], however, the complex remains in the A tubule of ependymal BBs even during ciliogenesis (Fig. 7), which may account for its divergent role of microtubule nucleation or anchoring during differentiation[63,64]. In a word, this intricate assembly is rigorously supervised and achieved through gradual protein remodeling and structural rearrangements.

For BBs, by applying subtomogram classification and averaging, the TMT structures were resolved at sub-nanometer resolution. Depending on the differential longitudinal localization of different TMT structures, BBs can be segregated into three distinct regions (Supplementary Fig. 2a), with the proximal region tethered to the cartwheel via a pinhead structure, while the central and distal regions are stabilized by a luminal inner scaffold. The latter two regions are distinguished based on the completeness of the C tubule (Fig. 7). In contrast to the traditional classification of BBs based on their association with the cartwheel, inner scaffold, and distal appendages[7,25], this study proposes a refined subdivision based on the inherent heterogeneity of the TMT structures themselves. This classification leverages features that directly correlate with the binding characteristics of TMTs to accessory compartments mentioned above (Fig. 2a–c). Structural analysis reveals that the redistribution of MAPs highly correlates with the progressive reduction of protofilaments within the C tubule in the distal region (Supplementary Fig. 2c, d).

Moreover, multiple WD40 domain-containing proteins are noteworthy for their significant contributions to the regulated transformation from proximal to distal BB and especially the transition of inner junction proteins (Fig. 2). The WD40 domain of POC1B is hypothesized to constitute the central core of inner junctions along BBs (Fig. 7). Nevertheless, within the inner scaffold region, this core likely interacts extensively with the WD40 domains from WDR90 to construct the foundational architecture of inner junction and inner scaffold. Our sub-nanometer structures promote a revision of the previously proposed model based on U-ExM[48]. This model suggested that the N-terminal DUF667 domain of WDR90, similar to CFAP20, participated in the joining of the A and B tubules within the inner scaffold region, while other WD40 domains resided in the lumen of TMT. This revision underscores the remarkable potential of in situ cryo-ET for elucidating the intricate architecture and composition of complex cellular machinery.

In addition, this study presents a detailed description of the annular structure composed of 27 rods, which emerges and constantly tracks the distal end of BBs after the BB assembly exceeds the cartwheel region (Figs. 4 and 7). The observation of identical structures in centrioles of HeLa cells[54,55] altogether suggests its conserved role in capping the distal central lumen. Analysis of neighboring microtubule structures suggests that it also acts as a reliable demarcation point for

the TMT-to-DMT transition. C2CD3 emerges as the potential candidate for this structure, as it interacts with the A tubule in the BB lumen of human RPE-1 cells and forms a ninefold-symmetrical ring with a radius roughly equal to that of the structure[41,65]. This assumption is further supported by its significant roles in the assembly of inner scaffold and distal appendages[66,67]. Future researches are necessary to elucidate the precise identity and function of the structure.

Structural analysis reveals that the remodeling of MIPs contributes to the transition from BBs to axonemes, as well as cilia maturation (Fig. 5 and Supplementary Fig. 5). In detail, different from the high occupancy of WD40 domains in the inner junctions within BBs, IJ$^{AB}$ of DMT$^{TZ}$ is solely composed of CFAP20 and the progressive incorporation of PACRG occurs beyond TZ as the axoneme extends (Fig. 7). Moreover, despite that the BB and TZ remain a consistent 8-nm periodicity, MIPs in the ciliary axonemes undergo a discernible remodeling to form the 48-nm repeat. However, due to the limited number of subtomograms and resolution, we cannot definitively exclude the possibility that other small MIPs in BB and TZ may exhibit higher periodicities to support the periodic changes. Compared to mature cilia, the structure of nascent cilia is characterized by the insufficient assembly of PACRG and the redundancy of two MIPs (mBMIP5 and mBMIP3L) (Figs. 5 and 7). mBMIP5 is located at B02/C02 along BBs and nascent axonemes while declining in mature axonemes. mBMIP3L alternates with NME7 to form a pseudo-8-nm periodicity at A09 protofilament in nascent cilia, akin to the 8-nm-repeat mBMIP3 in BBs and TZ (Figs. 3b, d, 5b and Supplementary Fig. 7). The loss of mBMIP5 and mBMIP3L highly correlates with a more advanced maturation stage of cilia (Fig. 5c). Derived from the similarities and discrepancies along BBs and axonemes, we assume that, during BB biogenesis and ciliogenesis, the microtubule assembly of a more distal region is initially templated from the relatively proximal region, followed by the selective remodeling of MIPs and MAPs to form distinct region-specific structures.

Using multiple approaches, we assigned the density of mBMIP5 to CEP41 and demonstrated its importance in multicilia formation (Fig. 6 and Supplementary Fig. 6). Here we identified CEP41 as a MIP located in B02 and C02 protofilaments of BBs and B02 protofilament of nascent axonemes, suggesting its potential role in the assembly and stabilization of B and C tubules (Fig. 7 and Supplementary Fig. 7). Consistently, CEP41 is absent from the C02 region in the distal BBs, where C01 protofilament is undergoing disassembly. Additionally, the distal or subdistal localization of CEP41 in long cilia coincides with the loss of mBMIP5 in mature cilia and exemplifies the constant turnover of tubulins in the distal flagella from *Chlamydomonas*[68], which remarkably makes it an early-stage binder to nascent cilia. However, the exact roles and functional disparities of CEP41 between BBs and cilia remain to be fully characterized. Previous studies have proposed that CEP41 contributes to ciliary tubulin glutamylation by recruiting the polyglutamylase TTLL6[58]. Its deficiency results in diminished GT335 labeling in mammalian primary cilia and zebrafish motile cilia[58,69]. It is intriguing that, although the glutamylation occurs on a single microfilament (B09) of the B tubule[70,71], the A tubule of pronephric motile cilia somehow becomes deformed in both CEP41 and TTLL6 zebrafish morphants[58]. Furthermore, tubulin is glutamylated on its C-terminal tail, which is positioned outside the microtubule lamina[70,71]. If TTLL6 is recruited into the lumen of the B tubule by CEP41 at B02, it is unlikely able to reach the C-terminal tails at B09 to glutamylate them. Future studies, including the generation of *CEP41*-deficient cell models and high-resolution structure of both TMTs and DMTs with CEP41 are thus required to clarify detailed functions of CEP41.

In conclusion, through in situ structural analysis, we have visualized several key snapshots during BB biogenesis and early ciliogenesis. Furthermore, detailed structural comparisons of the TMT and DMT states across various regions and stages have enabled us to propose a model of the well-regulated ciliary assembly.

## Methods

### Animals
Experiments involving mouse tissues were performed in accordance with protocols approved by the Institutional Animal Care and Use Committee of CAS Center for Excellence in Molecular Cell Science, Institute of Biochemistry and Cell Biology, Chinese Academy of Sciences. Wild-type mice (C57BL/6J, $n = 12$) at postnatal day 0 were employed for mEPCs culture and both females and males were used. A 2-month-old male mouse (C57BL/6J) was used to dissect the brain and testis tissues for immunoblotting.

### Cell line
HEK293T cells (ATCC) were cultured in high-glucose Dulbecco's modified Eagle's medium (DMEM, Thermo Fisher Scientific) supplemented with 10% (w/v) fetal bovine serum (FBS, GeminiBio) and maintained at 37 °C in 5% $CO_2$ incubator.

### Construct of the pLVX-GFP-Deup1 plasmid
The full-length cDNA of mouse Deup1 (GenBank accession no. KC211186) was cloned into the lentiviral expression vector (pLVX) with an N-terminal green fluorescent protein (GFP) tag. The CMV promoter was replaced by a 2-kb genomic DNA sequence upstream of the first exon of mouse Deup1 (Ensembl accession no. ENSMUSG00000039977).

### Lentivirus production
HEK293T cells were seeded into the 10-cm dish (Corning) one day before transfection. Cells in logarithmic growth phase were transfected with 10 μg lentiviral expression plasmids, 7.5 μg packaging plasmids (psPAX2) and 5 μg envelop plasmid (pMD2.G) by polyethylenimine transfection reagent (PEI, MW 40000, Polysciences, catalogue no. 24765-1) in a ratio of 1:3. The medium was replaced 8–12 h post transfection. After 48 h, the supernatant was collected and concentrated to 100 μL by Lentivirus Concentration Reagent (Biodragon, catalogue no. BF06205), and then stored in aliquots at −80 °C.

### mEPCs culture, transduction and transfection
The telencephalons were dissected from P0 mice (C57BL/6J), followed by removing cerebellum, olfactory bulbs, meninges and hippocampus in the cold dissection buffer (161 mM NaCl, 5 mM KCl, 1 mM MgSO₄, 3.7 mM CaCl₂, 5 mM HEPES, 5.5 mM Glucose, pH 7.4). The remaining tissues were digested at 37 °C for 30 min in fresh dissection buffer supplement with 0.2 mg/mL L-cysteine (Sigma, catalogue no. 30089), 0.5 mM EDTA, 1 mM CaCl₂, 1.5 mM NaOH, 10 U/mL papain (Worthington Bio Corp, catalogue no. LS003126), 5 U/mL DNase I (Sigma, catalogue no. D5025). The digestion was then stopped by adding 10% FBS. The cells were then gently pipetted and collected by centrifugation at $150 \times g$ for 5 min. The pelleted cells were resuspended in DMEM medium supplemented with 10% FBS, 4 mM L-glutamine (Sigma, catalogue no. 49419), penicillin-streptomycin (Thermo Fisher Scientific, catalogue no. 15140122) and 50 μg/mL Primocin (InvivoGen, catalogue no. ant-pm-05), and seeded into fibronectin (Sigma, catalogue no. FC010)-coated flasks. Neurons were shaken off over the next two days. The remaining cells were cultured for one more day and then seeded into fibronectin-coated 12-well plates (Corning). After reaching full confluence, cells were transduced with the concentrated lentivirus in a ratio of 1:20. Twenty-four hours post transfection, FBS was removed to induce the differentiation into multiciliated cells and the medium was changed every 2 days.

For the transfection of siRNA oligos, mEPCs reaching full confluence were cultured in a 29-mm glass-bottom dish (Cellvis, catalogue no. D29-14-1.5-N) and transfected with 40 pmol siRNA oligos and 3 μL Lipofectamine RNAiMAX transfection reagent (Thermo Fisher Scientific, catalogue no. 13778150) one day before serum starvation[39,72]. To improve the knockdown efficiency, the cells were transfected with the

siRNA oligos every three days. The sequences of siRNAs targeting *CEP41* (GenePharma) are listed below:

NC (Ctrli): 5′-TTCTCCGAACGTGTCACGTtt-3′;
siRNA #1: 5′-UAGACAAAGGGCUCGUAAAtt-3′;
siRNA #2: 5′-ACAAGAACGCCCACGGCAAtt-3′;
siRNA #3:5′-CAGAGACUGUUUCGGCAAAtt-3′;
siRNA #4: 5′-GUUGAAGAUUAAAGAACGUtt-3′.

## Cryo-ET sample preparation

mEPCs were harvested at day 3 post serum starvation and resuspended in DMEM with 10% glycerol before vitrification. The EM grids (Microgate film, Copper 200 mesh, XXBR) were pre-coated with an additional 20-nm-thick carbon layer using a high vacuum coating instrument (Leica EM ACE600) and glow discharged by a plasma cleaner (Gatan). Four microliters of the resuspended cells were applied to a grid in the Vitrobot chamber (FEI Vitrobot Mark IV, Thermo Fisher Scientific) with 90% humidity at 4 °C. The grids were blotted with filter paper (Whatman) on the back side and parafilm (Thermo Fisher Scientific) on the front side, and then plunged into liquid ethane. The vitrified grids were stored in liquid nitrogen until use.

## Cryo-correlative light and electron microscopy and cryo-FIB milling

The grid was assembled into AutoGrid and then loaded onto the cryo-stage of the Thunder Imager EM Cryo CLEM (Leica) equipped with a 50×/0.9 NA objective. The overview images of the grid and the GFP signal were acquired through the reflection channel and the corresponding fluorescence channel, respectively. Overlay images were exported to FEI Maps software to correlate with the SEM images, which guides the focused-ion beam (FIB) milling. Briefly, the grids were transferred into the Aquilos2 cryo-FIB/scanning electron microscope (SEM, Thermo Fisher Scientific). To reduce the charging effect, a protective layer of organometallic platinum was deposited by the gas injection system (GIS), followed by the sputter coating. After the alignment between the fluorescence image and SEM image, rough milling was performed with a current of 0.5 nA in the fluorescence-containing region to produce an initial 1.5-µm-thick lamella. The current was then progressively reduced and the 30 pA current was used for final polishing, resulting in a lamella with a thickness of 150−200 nm. The lamellae were stored in liquid nitrogen for subsequent data collection.

## Cryo-ET data collection

The lamellae were transferred into a 300 kV Titan Krios electron Microscope (Thermo Fischer Scientific) equipped with a K3 direct detector camera (Gatan). Tilt series were collected using SerialEM[73] with the Plugin PACE-tomo[74] from +66° to −38° with a 2° increment (beginning at a pretilt of +14°) and the dose-symmetric tilt scheme was applied[75]. The magnification of record was set to 26,000× with the corresponding pixel size of 3.328 Å/pixel, and the defocus range was −3 µm to −5 µm. Images were recorded in a super-resolution mode with 10 frames per image and the total dose was limited to -110 e−/Å² for each tilt series. 261 tomograms were acquired in total from six individual cryo-ET sample preparations.

## Tomogram reconstruction and segmentation

Initially, the preprocessing of the collected tilt series was conducted in TOMOMAN software[76]. The frames from each tilt angle underwent motion correction through MotionCor2 software[77] to generate tilt series, which were imported into IMOD software[78] to perform patch-tracking-based alignment and reconstruction. For particle picking and visualization purposes, tomograms were downscaled via a binning factor of 8 and then deconvolved with MATLAB TOM toolbox[79]. The images of tomogram slices were displayed in IMOD software.

The binned tomograms were missing-wedge-corrected using IsoNet[80]. Membranes were segmented using MemBrain-Seg (https://github.com/teamtomo/membrain-seg) and then manually optimized using Amira (Thermo Fisher Scientific). For the annotation of cilia, the density maps, coordinates and Euler angles of TMT and DMT generated from particle picking and STA were utilized. The visualization and movie of segmentation were conducted with UCSF ChimeraX[81].

## Particle picking and subtomogram averaging

For TMT, the centers of basal bodies were manually picked from proximal to distal ends in the IMOD software. Particles were oversampled at 4 nm intervals through the 'relion_helix_toolbox' program in RELION2[82]. The subtomograms were extracted at the pixel size of 26.624 Å/pixel in Warp[83] and aligned to a cylindrical reference with C9 symmetry in RELION2. Symmetry expansion on the average map of basal body produced TMT particles and the subtomograms were initially extracted at the pixel size of 26.624 Å/pixel for 3D refinement. To exclude duplicated particles, the particles that were within 4 nm of their nearest neighbors were removed. Subsequently, manual inspection, primarily focusing on the *Rot* angle in RELION STAR files, was performed utilizing the ArtiaX plugin in ChimeraX to discard the misaligned particles. The coordinates of the yielding 33,516 particles were shifted to the center of A and B tubules and the subtomograms were re-extracted at the pixel size of 3.328 Å/pixel. Further 3D refinement with a focused mask on A and B tubules was performed to remove misaligned particles, yielding 29,544 particles. The result was then refined in the software M[84], producing the 8-nm-repeat TMT density map with only A and B tubules (TMT^AB) at a global resolution of 7.6 Å. 3D classification with a local cylindrical mask on the pinhead produced two subsets of 6538 particles with pinhead (TMT^PH) and 20,849 particles with inner scaffold (TMT^IS). The particle coordinates were then shifted to the center of B tubule. 3D classification on TMT^IS with a local cylindrical mask on C tubule, resulted in two subsets of 11,794 particles with complete C tubule (TMT^IS_comC) and 7016 particles with incomplete C tubule (TMT^IS_incomC). After further 3D refinement, the final maps have an indicated global resolution of 9.1 Å for TMT^PH, 8.4 Å for TMT^IS, 8.8 Å for TMT^IS_comC and 9.2 Å for TMT^IS_incomC at a Fourier shell correlation of 0.143.

For doublet microtubules (DMT), the centers of axonemes were manually picked from proximal to distal ends in the IMOD software. Subsequently, particles were sampled at 8 nm intervals through the 'relion_helix_toolbox' program in RELION2. A total of 32,322 particles from 67 tomograms were extracted using the WARP software with a pixel size of 6.656 Å/pixel and a box size of 96 pixels. Data processing was performed with RELION2. 3D refinement was carried out with two Euler angles (tilt and psi) restricted around prior values. The results were then manually checked using ArtiaX plugin in ChimeraX, and the incorrectly aligned particles were removed. The remaining 28,793 particles were then cropped in WARP with a pixel size of 3.328 Å/pixel and a box size of 192 pixels for further 3D refinement with a local search range. This result was then refined in the software M, yielding the 8-nm-repeat DMT density map (DMT^8nm) with a global resolution of 10.9 Å, as well as improving the alignment of tilt stacks. The particles were cropped again with a pixel size of 6.656 Å/pixel and a box size of 96 pixels. 3D classification with a cylindrical mask near protofilaments B09-B10 (focused on the 16-nm-repeat CFAP52) produced two subsets of 13,070 particles with 16-nm repeat (DMT^16nm) and 2176 particles mainly distributed in the transition zone (DMT^TZ). The particles of DMT^16nm were subjected to subsequent 3D classification with a cylindrical mask near protofilaments A09-A10 (focused on the 48-nm-repeat CFAP52 and NME7), yielding the 48-nm-repeat DMT density map (DMT^48nm). The particles of DMT^TZ, DMT^16nm and DMT^48nm density maps were re-extracted with a pixel size of 3.328 Å/pixel and a box size of 192 pixels for final 3D refinement. The final maps have an indicated

global resolution of 12.6 Å for DMT$^{TZ}$, 10.8 Å for DMT$^{16nm}$ and 14.2 Å for DMT$^{48nm}$ at a Fourier shell correlation of 0.143.

Local resolution of the density maps was determined in RELION and surface coloring was performed using UCSF ChimeraX.

### The 27 rods

The A tubules closest to the 27 rods were manually picked, resulting in 242 particles. The particles were cropped in WARP with a pixel size of 6.656 Å/pixel and a box size of 168 pixels. 3D refinement was then performed with a reference of cylinder to get an initial structure of TMT and the connecting rods. To achieve a finer structure, two additional rounds of alignment were performed in RELION2, each applying a different mask. One contained the TMT and the connected T-shape density, while the other focused on the rods and the adjacent T-shape density.

### 2D projection

After the STA of TMT and DMT, the coordinates and Euler angles of all particles were determined. Particle coordinates were shifted to the target MIPs (e.g., NME7) and then cropped from tomograms using WARP with a pixel size of 6.656 Å/pixel and a box size of 192 pixels. Subsequently, the cropped particles were rotated to a consistent orientation based on the Euler angles and projected along the z-axis to generate 2D projections for the central two slices. Four examples of 2D projections were shown for each MIP (Supplementary Fig. 7) in the basal body, transition zone and axoneme beyond the transition zone.

### Model building and CEP41 identification

For map interpretation, the atomic model of the mouse sperm DMT (PDB accession: 8IYJ)[12] was fitted into the density maps using UCSF ChimeraX and manually corrected in COOT[85].

For mBMIP5, the coordinates from refined TMT$^{AB}$ were shifted to the center of mBMIP5 density at B01 region, and the coordinates from refined TMT$^{PH}$ and TMT$^{IS\_comC}$ were shifted to the center of mBMIP5 density at C01 region. Subtomograms were then extracted using all these shifted coordinates at the pixel size of 3.328 Å/pixel and with a box size of 56 pixels, yielding 47,876 particles. 3D classification with local angle search and a soft mask around mBMIP5 was executed to exclude any poor reconstruction. Further 3D refinement of resting 44,672 particles produced the final density map with a global resolution of 8.1 Å at a Fourier shell correlation of 0.143. To identify mBMIP5, the corresponding density at the B02 region in the B tubule was cropped from the TMT map using UCSF Chimera[86]. The AlphaFold2 predicted models of mouse proteome were automatically fitted into the density by Colores program of the Situs package[15,52,87,88]. The fitted models were ranked by cross-correlation scores and the top hits were manually checked in UCSF Chimera. This procedure was repeated by searching a smaller library of proteins under 45 kDa. The best match was CEP41 for both experiments. The AlphaFold2[51,52] predicted CEP41 model was refined against the density map using COOT.

### Immunoblotting

For immunoblotting, day-9 mEPCs were harvested to verify the knockdown efficiency of the CEP41 RNAi experiments (Fig. 6h). Day-10 mEPCs and day-2 IMCD3 cells (ATCC) were used to compare the CEP41 expression levels in motile and primary cilia (Supplementary Fig. 6c). Specifically, IMCD3 cells were cultured for 2 days post serum starvation to stimulate the formation of primary cilia. Brain and testis tissues were excised from a 2-month-old male mouse.

The samples were lysed in the 2× SDS-loading buffer and boiled at 100 °C for 8 min. Proteins were then separated by SDS-PAGE using Precast 4.5-10% Bis-Tris gels (Epizyme Biotech, catalogue no. PG212) and transferred onto the nitrocellulose membrane with a pore size of 0.2 μm (GE Healthcare, catalogue no. 10600001). The membrane was then washed three times in TBST (150 mM NaCl, 50 mM Tris-HCl,

0.05% Tween-20, pH 7.5), and blocked with 5% non-fat milk in TBST for 1 h at room temperature. Subsequently, it was incubated overnight at 4 °C with the following primary antibodies: rabbit anti-CEP41 (Affinity Biosciences, 1:2000, catalogue no. DF9362, lot no. 89H4950) and rabbit anti-Gapdh (Proteintech, 1:5000, catalogue no. 10494-1-AP, lot no. 00103483) in TBST with 1% BSA. The membrane was then washed three times in TBST for 15 min each and incubated with HRP-conjugated goat anti-rabbit secondary antibodies (Thermo Fisher Scientific, 1:5000, catalogue no. G-21234, lot no. 2156243) in TBST with 1% BSA for 1 h at room temperature. After three 15-min washes, proteins were detected using Western Lightning Plus-ECL (PerkinElmer, catalogue no. NEL104001EA), and the luminescent signals were captured with the MiniChemi Chemiluminescence imager (Sagecreation).

### Immunofluorescent staining and microscopy

mEPCs, cultured in the 29-mm glass-bottom dishes (Cellvis), were treated with 0.5% Triton X-100 in PBS for 30 s, followed by the fixation with 4% fresh paraformaldehyde (PFA) in PBS for 15 min at room temperature. The cells were then permeabilized with 0.5% Triton X-100 in PBS for 15 min and blocked with the blocking buffer (4% BSA in TBST) for one hour at room temperature. The samples were subsequently incubated overnight at 4 °C with the following primary antibodies in the blocking buffer: rabbit anti-CEP41 (Affinity Biosciences, 1:1000, catalogue no. DF9362, lot no. 89H4950), mouse anti-acetylated-tubulin (Sigma, 1:1000, catalogue no. T6793, lot no. 0000269851), guinea pig anti-Odf2 (home-made, 1:500), followed by three washes with the blocking buffer for 5 min each. The samples were then incubated with DAPI (Sigma-Aldrich, 1:1000, catalogue no. D8417) and the following secondary antibodies in the same buffer: Alexa Fluor 647-conjugated goat anti-rabbit (Thermo Fisher Scientific, 1:1000, catalogue no. A-21245, lot no. 2442141), Cy$^{TM}$3-conjugated donkey anti-mouse (Jackson ImmunoResearch, 1:1000, catalogue no. 715-165-151, lot no. 162530), Alexa Fluor 488-conjugated donkey anti-guinea pig (Jackson ImmunoResearch, 1:1000, catalogue no. 706-545-148, lot no. 158591), for one hour at room temperature. After three 5-min washes with the blocking buffer, the samples were mounted using ProLong$^{TM}$ (Thermo Fisher Scientific, catalogue no. 2273640). Confocal microscopy images were captured using the Leica TCS SP8 WILL system, equipped with a ×63/1.40 oil immersion objective. Additionally, Z-stack images were obtained through maximum intensity projections.

### Live cell imaging

For the live imaging of day-3 and day-10 mEPCs, cells cultured in the 29-mm glass-bottom dishes (Cellvis) were stained with 200 nM SiR-tubulin (Spirochrome, catalogue no. SC002) and 10 μM Hoechst 33342 (Meilun, catalogue no. MB3210) for one hour at 37 °C in 5% CO$_2$ incubator to label microtubules and nuclei, respectively. The cells were covered with a 12-mm coverslip on the cell surface and images were captured by confocal laser scanning microscopy (Leica TCS SP8 WILL), equipped with a ×63/1.40 oil objective.

For live imaging of ependymal ciliary motions, imaging was performed at the intervals of 60 ms (day-3 cilia) or 50 ms (day-10 cilia) using a spinning disk confocal microscope (Olympus SpinSR), equipped with a UPlanXApo ×60/1.42 oil objective. The imaging was acquired with an ORCA-Flash 4.0 V3 digital CMOS camera (supplied by Hamamatsu) and the OBIS solid state lasers was set at 650 nm with 80% laser power. The cells were maintained in an incubation chamber at 37 °C with 5% CO$_2$ and 80% humidity. The time-lapse images were processed and analyzed using Cellsens Dimension software (Olympus) and Fiji (a distribution of ImageJ). Three recognizable cilia of each cell were analyzed.

### Quantification and statistical analysis

For cryo-ET data processing, a total of 261 tomograms were collected from six independent sample preparations. For microscopic and

biochemical findings, they were replicated a minimum of twice to ensure accuracy. The percentage of multiciliated cells (Figs. 1c and 6j) was determined by counting the number of DAPI/Hoechst-positive nuclei and acetylated-tubulin/SiR-tubulin-positive cells visible in confocal images. The ciliary lengths of multiciliated cells (Figs. 1d and 6e) in the micrographs were precisely measured using ImageJ software and divided into three regions (long, medium, and short) based on the equal distance scatter method. The average length of the three longest cilia within a multiciliated cell was calculated to represent the length of the multicilia. Unless otherwise stated, the quantification results are presented as the mean ± SD. Two-sided student's $t$-test was used to calculate the $P$-values between unpaired samples in the GraphPad Prism software. Differences were considered statistically significant when $P < 0.05$. Only data obtained from three or more independent experiments were included in the $t$-tests. All the details can be found in the figure legends and method details.

### Reporting summary

Further information on research design is available in the Nature Portfolio Reporting Summary linked to this article.

## Data availability

The 48-nm repeat DMT structure from day-10 mEPCs[19] are available with EMD-37111. The tomographic reconstruction of the centriole in HeLa cells can be found with EMD-33495. The 96-nm repeat of human respiratory DMT complex[13] can be accessed with EMD-35888. The DMT structures from bovine tracheal cilia[16], mouse sperm flagella[12], human respiratory cilia[13] and sea urchin sperm flagella[14] are available with 7RRO, 8IYJ, 8J07 and 8SNB, respectively. The genomic DNA sequence of Deup1 can be accessed at [https://www.ncbi.nlm.nih.gov/nuccore/KC211186.1/] and [https://www.ncbi.nlm.nih.gov/gene/?term=ENSMUSG00000039977]. The subtomogram averaging maps from mEPCs in the study have been deposited in the Electron Microscopy Data Bank (EMDB) with the following accession numbers: EMD-39652 (8-nm repeat TMT[AB]), EMD-39654 (8-nm repeat TMT[PH]), EMD-39653 (8-nm repeat TMT[IS_comC]), EMD-39655 (8-nm repeat TMT[IS_incomC]), EMD-39656 (8-nm repeat DMT[TZ]), EMD-39657 (48-nm repeat DMT[48nm]), EMD-39658 (8-nm repeat CEP41). The tomograms analyzed in Fig. 1 have been deposited at the EMDB with accession numbers: EMD-39659 (Fig. 1g) and EMD-39660 (Fig. 1e and Supplementary Movie 1). To achieve broader accessibility, more tomograms depicting the deuterosome-dependent basal body biogenesis at various stages have been deposited at the EMDB with accession numbers: EMD-63282, EMD-63283, EMD-63284, EMD-63285, EMD-63286, EMD-63287, EMD-63288 and EMD-63289. All data supporting the results of this study are provided in the article, supplementary information, and source data file. Source data are provided with this paper.

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

## Acknowledgements

We thank Prof. Ning Gao, Prof. Chuanmao Zhang, and Prof. Jianguo Chen at Peking University for their critical suggestions. We thank Chen Chen at Peking University for the assistance in mEPCs culture. We are grateful to the Cryo-EM Platform of Peking University and Changping Laboratory, the High-Performance Computing Platform of Peking University, the Cryo-Electron Microscopy Facility of Liangzhu Laboratory at Zhejiang University for the support on data collection and computation, the National Centre for Protein Sciences and the Core Facilities at the School of Life Sciences at Peking University and the Core Facility of Liangzhu Laboratory at Zhejiang University for technical assistance. This work is funded by the Beijing Natural Science Foundation (JQ24031 to Q.G.), the National Natural Science Foundation of China (#32230027 to X.Z., #32371191 to Q.G., #32471245 to M.G.), the National Key Research and Development Program of China (#2022YFC2303700 to Q.G.), the Excellent Youth Science Fund (Overseas) to M.G., the Zhejiang Provincial Natural Science Foundation (LZ24C050001 to M.G.), and the Key R&D Program of Zhejiang (No.2024SSYS0022 to M.G.). Q.G. is supported by Changping Laboratory, the SLS-Qidong Innovation Fund and the Li Ge-Zhao Ning Life Science Youth Research Foundation.

## Author contributions

Q.G. conceived the project. S.M. cultured mEPCs, performed the electron microscopy experiments and collected cryo-ET datasets. B.Q. supported the mEPCs culture. W.D. assisted in the electron microscopy experiments and data collection. S.M., Z.L. and S.L. performed cryo-ET data analysis. L.L. cultured cells and performed the functional experiments and fluorescent microscopy. Q.L. and M.G. conducted the model refinement and protein identification. S.M., Q.G., X.Z and M.G. wrote the manuscript with the input from all authors.

## Competing interests

The authors declare no competing interests.
