## [Transparent Peer Review file · Nature Communications]

In situ cryo-electron tomography reveals the progressive biogenesis of basal bodies and cilia in mouse ependymal cells

Corresponding Author: Professor Qiang Guo

Version 0:

Reviewer comments:

Reviewer #1

(Remarks to the Author)

Ma and coworkers provide a gorgeous example of how start-of-the-art structural techniques can resolve architectural features of biological machinery. By creating lamella from ciliated mouse ependymal cells (mEPCs) expressing GFP-labeled Deup1, they are able to use cryo-ET to visualize deuterosomes, basal bodies, and the proximal regions of axonemes. Using subtomogram averaging, they resolve subnanometer resolution reconstructions of the triplet microtubules (TMTs) of the basal body and the doublet microtubules (DMTs) of the transition zone and axoneme. These reconstructions improve on previous ex-vivo structures of bovine basal body-axoneme complexes but fall short of being able to identify most of the microtubule-bound proteins. Regardless of this limitation, the paper represents an impressive advance that should be published once the following issues have been addressed:

Major comments

1. Biogenesis of basal bodies and cilia may vary in different organisms/cell types; the title should therefore reference that this study has been conducted in murine ependymal cells
2. The paper needs a table summarizing the different reconstructions; their origin (e.g. day-3 or day-10 mEPCs), their applied periodicity, particle number, and resolution. Currently, this information is spread throughout the text.
3. In bovine samples, the DMT TZ is clearly demarcated with shelf-like MIPs in the A and B tubules (Greenan et al, 2019 JCB). These appear to be absent from the reconstructions presented here. Does this represent a difference between species, or a result of imaging “young” axonemes? Without this internal density, is the DMT TZ a distinct type of DMT, or simply a gradual transition between TMTs and DMTs? Exactly what is being imaged, and how this might relate to other species should be addressed.
4. The presence of densities consistent with axonemal MIPs (e.g. MNS1, CFAP53, NME7, CFAP141, and ENKUR) in the “DMT TZ” map suggests that 8 nm periodicity is being incorrectly applied and the true periodicity is higher.
5. More information should be provided on the deuterosomes. The central hub (CH) is a prominent feature of the tomograms but is not discussed in the paper. Is a reconstruction possible?
6. Identifying proteins at 8 Å is challenging, although the identification of CEP41 appears convincing and is supported by additional evidence. A reconstruction of MTs from the CEP41 knockdown cells demonstrating loss of B tubules or CEP41 density would strongly support the conclusion. If this is not possible, tomography of the cells following knockdown should be done to show the gross morphological changes.
7. To test the model that CEP41 enhances microtubule glutamylation, the authors should test whether tubulin glutamylation is reduced in the CEP41 knockdown mEPCs.
8. The authors compare reconstructions of DMTs from day-3 and day-10 mEPCs to conclude that two MIPs are lost during maturation: mBMIP3L and mBMIP5. However, it is unclear if the reconstructions come from the same region of the axoneme. Are these differences a consequence of different stages of biogenesis or reconstructing more distal DMTs from the later time point?
9. Were TMT structures also compared from day-3 and day-10 mEPCs to show that they have the same MIP/MAP composition? This information should be included.
10. Both day-3 and day-10 can be considered “early” in the life cycle of cilia. The study would be enhanced by the addition of a much later time point. Does TMT/DMT structure change further?
11. Movies should be provided showing the quality of the density and, in the case of CEP41, the quality of the fit to density.

Minor comments

- It should be made clear that the highest reported resolution (7.6 Å) is not biologically meaningful as it corresponds to a mixture of TMT identities and is only for the A and B tubules.
- The authors' statement that all BB MIPs/MAPs have 8-nm periodicity should be qualified because smaller densities which may have higher periodicities cannot be resolved at 8-9 Å resolution.
- The authors suggest that they observe "strands of membrane particles that appear to anchor the ends of Y links" but also mention that they do not observe Y links. This inconsistency should be explained. Why are Y links not observed? Are proteins thought to be present in Y links expressed in mouse ependymal cells?
- The current results section contains speculation that is better suited to the discussion, e.g. "striations may contribute to the stiffness and compressibility of the TZ axoneme".
- The authors state that both intra- and extracellular pathways are required for efficient multiciliogenesis but do not show this experimentally. Without knowledge of the referenced papers, it is unclear how their observations revise current models of multiciliogenesis. Further explanation of the current models would be helpful to understand the implications of the new observations.
- Even in day-3 mEPCs, long cilia exist. It is possible to determine from the lamella the cilia length of the BBs being imaged?
- PMID: 38841887 must be cited as it provides supporting evidence that CEP41 is a microtubule-binding protein.
- A C2CD3 knockdown, followed by tomography to determine if rods were lost would enhance the study, but is not essential for the revision.
- Please provide additional information on how duplicated particles from oversampling were excluded, as duplicated particles can elevate resolution estimates.
- Please clarify which particles were used to generate the 8.1 Å resolution map used to identify CEP41. Are these particles coming from different classes of microtubules?

Reviewer #2

(Remarks to the Author)

In this manuscript, the authors utilized cryo-electron tomography to investigate the structures of ependymal cilium basal bodies and the microtubule doublets of nascent and mature ependymal cilia to clarify the assembly process of ependymal cilia. The specimens were prepared with cryo-FIB under the guidance of fluorescent imaging, which provided the authors with 177 useful tomograms for the structural analysis. The density maps after sub-volume analysis are of excellent quality, enabling the authors to determine the detailed structural differences of the basal bodies at the proximal, central, and distal regions and identify the doughnut-shaped densities as WD40 combining proteins, an 8-nm-repeat microtubule-associated protein in the lumen and a barrel of 27 rods at the border between the doublet and triplet microtubules. RNAi was also carried out in cooperation with cryo-ET, which enabled the determination of the identity of the mBMIP5 density to be CEP41. Overall, this manuscript represents a significant body of excellent experimental works that combine the high-resolution structural determination of the ependymal cilia basal body with detailed structural information of microtubule doublets in ependymal cilia at different maturation stages, as well as live cell imaging analysis of the ependymal cilia at various maturation stage. Besides providing detailed structural information on the ependymal basal body, the manuscript offers insight into the assembly process of microtubule doublets at the structural level. I think the research work in this manuscript is a very strong candidate for publication in Nature Communication.

I have the following concerns for the authors to consider:

Major concern:

The manuscript could profit from some reorganization, revision of statements, and additional background in the Introduction. In its current form, each part of the content is very good and relatively clearly written. However, these parts are not well-organized and linked, making it difficult for readers to appreciate the merit of the outstanding experimental works.

The manuscript's title emphasizes the "progressive biogenesis of basal bodies and cilia". The first part of the Results describes the strategy of establishing a pipeline that captures various stages of ciliogenesis in mEPCs. Based on this section, Day 3, after serum starvation, provides multi-ciliated cells at various differentiation stages, including massive basal body biogenesis, while on Day 10, the majority shows long cilia. After this section, bearing the manuscript title in mind, readers would expect results of basal bodies of various stages (from I to IV? May be too much; At least several stages) to understand basal body biogenesis. However, the Results section only provided one complete structure of the basal body and its associated proteins. Indeed, the structural work of the basal body was carried out nicely, and the density map is excellent. It revealed structural differences at the basal body's proximal, central, and distal portions. Some microtubule-associated proteins, including the barrel of 27 sticks, were identified through convincing density map analysis. These are excellent structural information that represents a great deal of effort. However, there are six stages of the basal body genesis (Page 4, line 106, and Figure 1a and 1b); which stage of the basal body provided this cryo-ET map? The manuscript failed to provide this critical information in the context of basal body genesis. There is a significant disconnection of the logic thread for the whole manuscript. I am not requesting several basal body structures at different assembly stages since it may require significantly more work than practical. However, the current design and some statements in the manuscript lead to high expectations.

The basal body structure in the manuscript seems to be obtained with the mature basal bodies (Day 10). Let's assume so. How the single snapshot structural result, although an excellent structural snapshot, provides insight into progressive basal body genesis has yet to be established. The basis of the hypothesis proposed in Figure 7 is not clarified. It seems the

manuscript assumes the structural difference from the proximal end to the distal end of the basal body represents the assembly process of the basal body. While it is possibly true, when no direct data backs up, there lacks a logical link between the structural data of the basal body and the basal body genesis model proposed in Figure 7. If so, the title of the manuscript is inappropriate, and some of the statements appear exaggerated, damaging the merit of this excellent research work. The manuscript should be revised with a better design. Suppose there is some cryo-ET data for the assembling basal bodies before they mature, as implied by a few inserts in Figure 7. In that case, the supporting data should be organized into a figure compared to the structural results of the mature basal body to support the hypothesis that the structural difference between the proximal and the distal region represents the different stages of the basal body assembly process.

The manuscript appears to be trying to capitalize on the research design for the structural investigation of ependymal ciliogenesis to strengthen its biological significance. While the microtubule doublet part fits the design, the basal body part, although excellent and containing abundant new information, is disconnected. The authors may consider focusing the cilia genesis study design on the microtubule doublet work and making a separate research justification for the basal body structural work. If so, Fig 7 is no longer essential. There is sufficient new structural information in the basal body structural work. This is not a request but a suggestion.

The introduction part is too structure-focused. To capitalize on the biological insights obtained through the structural study in the manuscript, more background on ependymal cilia development, e.g., deuterosome-dependent multi-cilia genesis, should be added.

Minor concern:

(1) Page 21 Line 524, Figure 3 legend: “mBMIP1 resembles MNS1 in cilia;” It is a confusing statement. How about “mBMIP1 resembles MNS1 in doublet microtubule;” ?

(2) Figure 5. The overall summary of the figure states, “Nascent cilia revealed by in situ cryo-ET,” which is inappropriate. Panel 5d of the figure has nothing to do with the figure summary and it contains data for both nascent cilia on Day 3 and natural data from Day 10.

In fact, the structural comparison of the nascent (Day 3) and mature (Day 10) microtubule doublets represents a significant result in this manuscript, as it provides insights into the assembly process of microtubule doublets. The results in the Extended Information Figure 5, at least the S-5b, should be included in the main figures. For example, the S-5b may be included in current Figure 5. If so, the summary of Figure 5 may change to “Difference between the nascent and mature microtubule doublets”.

(3) The extended data Fig 3 demonstrates basal body template ciliogenesis, showing both extracellular and intracellular pathways. This is one of the key biological insights this manuscript provides. I suggest the author consider moving it to the main figures if space allows. Meanwhile, background information about extracellular and intracellular pathways of the basal body assembly regulation should be provided in the Introduction part.

(4) Page 35, Line 787, “ For immunoblotting, day-10 mEPCs (day-9 mEPCs for RNAi) were harvested. IMCD3 cells 787 were cultured for 2 days post serum starvation to stimulate the formation of primary cilia.” Did the authors introduce this part from other manuscripts but forget to modify?

Reviewer #3

(Remarks to the Author)

The authors of the manuscript “In situ cryo-electron tomography reveals the progressive biogenesis of basal bodies and cilia” establish a workflow to analyse structural details of ciliogenesis by employing a cryo-CLEM, cryo-FIB and cryo-ET workflow. To this end, myeloid endothelial progenitor cells (mEPCs) isolated from mouse brain were utilized as a model system for multiciliated cells and transduced with a Deup1-GFP fusion to enable targeted lamellae preparation. Cryo-ET data collection and subsequent subtomogram averaging allowed structural analysis of basal bodies (BB) in situ. Interestingly, tomograms revealed that mEPCs utilize both the intracellular and extracellular pathway for ciliogenesis, different from what was previously thought.

Based on the presence of certain structural features three different BB triplet microtubule (TMT) regions were defined for STA analysis and resolved to 9.1 Å (TMTPH), 8.8 Å (TMTIS_comC) and 9.2 Å (TMTIS_incomC). The subtomogram averages reveal common structural features of centriolar/BB TMTs like the pinhead and A-C linker, and variation in MIP decoration along the three different regions are observed. While the achieved resolution does not allow clear identification and assignment of MIPs and MAPs, several WD40 domains are identified to be closely associated with TMTs and speculated to be instrumental to enhance stability and integrity of BB TMTs. Based on STA combined with fitting of AF2 models the authors identify mBMIP5 to be CEP41. This is further supported by knock-down assays with CEP41-specific siRNA in mEPCs. STA of the transition zone doublet microtubules (DMTTZ) at a resolution of 12.1 Å showed a similar pattern of MIP decoration as the TMTIS_incomC region, however differences are observed at the inner junction. Furthermore, a distinct structure marking the transition from TMT to DMT, termed as rods, is described as a boundary to the TZ. Finally, the authors compile their results into a model that describes the different steps of ciliogenesis and highlight how MIP decoration evolves throughout the different regions.

The presented study provides some novel insight into ciliogenesis in situ. Although many of the structural features identified have been described before, the comprehensive nature of the study, covering the structural features along entire BB and

providing an overview of these will be of significant value to the ciliogenesis field. Similarly, the observation that both extra- and intracellular ciliogenesis pathways appear in multiciliated cells advances insights into ciliogenesis pathways employed by different cell types. However, the reviewer would like to add that ciliogenesis is not their core expertise. Since the resolution of the different STAs is not sufficient to clearly identify individual proteins, the significance of the structural insights on the BB TMT and DMT regions seems more limited. Yet, the study investigates these structures in a cell type previously not studied in that much detail and importantly from a metazoan, thus not only supporting previous STA studies but further extending our knowledge on how conserved these structures are.

The work largely supports the claims made. The data analysis and interpretation are mostly coherent. The conclusions drawn do sometimes seem to not be fully supported by the data. Thus, some clarification is required to ensure some of the data is not overinterpreted by the authors (see comments below).

The described methodology is sound and meets the expected standard within the field.

The material and methods are mostly described in enough detail to reproduce the work. A few more details are required as indicated below.

Major

The reviewer is surprised that, while the central hub of the cartwheel is clearly visible in the tomograms, the spokes of the cartwheel do not seem to be present. As the spokes of the cartwheel are part of the final model (Figure 7), absence of the cartwheel spokes should at least be discussed.

The authors define TMTIS_incomC as a category of TMT before the transition zone. While previous studies and some of the raw data presented here (Extended data Figure 4 a, b and d) suggest a gradual change from a full C-tubule to a missing C-tubule, the STA presented in figure 2A suggests a clear class of 6 PF C-tubule in that region. It should be discussed in the main text how sharp this transition really is (meaning: is it really full C-tubule, 6PF C-tubule, no C-tubule).

The authors state the observation that BBs template ciliogenesis via extra- and intracellular pathways. Since the field previously expected multiciliated cells to prefer the extracellular pathway, it would be great to substantiate the authors claim with a quantitative comparison. The size of the dataset should allow some statistical analysis.

It is not clear to the reviewer, which data exactly (TMT or DMT, which part of TMT/DMT?) was used and refined to identify CEP41. This needs to be more clearly specified either in the corresponding methods section or in the main text and corresponding figure. Furthermore, at the present resolution the reviewer would suggest a more conservative interpretation of the identification. The authors claim that based on cross-correlation CEP41 is the most probable candidate, but clear identification seems not fully supported by the average.

The siRNA experiment particularly quantifies cilia formation. It would be important to also quantify another baseline parameter, like viability, as a control and to show that CEP41 knock-down does not affect other important cellular functions.

Line 304-310: In this paragraph, DMT structures from nascent and mature DMT48nm are compared. It is not completely clear to me whether nascent automatically means that the structures come from 3-day mEPCs and mature from 10-day mEPCs. Furthermore, it seems to me that this is the first time averaging of 10-day mEPCs appears and that in the material and methods only 3-day mEPCs are mentioned to be used for preparation of grids. Where is the average of the 10-day mEPCs coming from? Why is it only shown in the extended figure? To allow a reader an easy comparison, as described in the text, it would be nice to have a panel in the main figure. Also, the reviewer is wondering whether radial spokes, IDA or ODA were more clearly visible in averages from the mature 10-day mEPCs

Minor

The density in cyan correlated to POC1B at IJAB (Figure 2a,b,c,d,f) seems to stretch in two directions, towards the A and the B tubule. However, when fitting POC1B a stretch to only one side is fitted with the extending α -helix of POC1B. Does the other extended density not belong to POC1B? If so, why is it the same colour?

In a few instances, the manuscript would profit from more clarity in writing and a more coherent connection between the figures and different structural features highlighted in the text. A (non-exhaustive) list of examples and a few typos can be found below:

Line 86: "larger than" instead of "large than"?

Line 125: "the structure" refers to the cartwheel?

Line 256-260: First, periodicities and identities of MIPs of DMTTZ is said to be like TMT, except for inner junction. But in the next sentence the DMTTZ inner junction is compared to ciliary axoneme DMTs. What is the difference in the IJ between TMT and DMTTZ? Then afterwards it can still be compared to ciliary axoneme. The authors could also highlight this difference in the according figure (figure 3).

Line 270: "in another word" should be "in other words"

Line 442-445: It is not clear to me what this sentence aims at. Rephrasing it would definitely help to make it clearer.

Figure 6: what does the cyan colour indicate? Or is the red colour just indicating the RHOD and the cyan the "not-RHOD"? It would be nice to indicate into which direction disordered loops would continue (if possible without making the figure visually incomprehensible).

Extended data Figure 4 a: sentence in line 974 and 975 is a bit unclear. The white box in the figure is hard to see.

Version 1:

Reviewer comments:

Reviewer #1

(Remarks to the Author)

I thank the authors for their thoughtful responses to my comments and questions. I also appreciate their efforts to investigate the relationship between CEP41 and tubulin glutamylation and agree that the preliminary results are beyond the scope of the paper and the question is best addressed in future investigations. Overall, the changes and additions have improved the paper, and I now recommend it unreservedly for publication.

Minor comments:

L84. The newly introduced sentence: "Deup1 (deuterosome protein 1, also known as Ccdc67), a paralogue of Cep63, which is a mother centriole-localized, Cep152-interacting protein important for centriole biogenesis, forms the core of deuterosomes" would benefit from simplification.

L100. "High-resolution" and "entire" should be removed from this sentence.

L170 "The protofilaments of C tubule in TMTIS_comC was gradually shedding from C10 during proximal to distal BBs (Extended Data Fig. 2c)." This sentence could be revised for clarity.

Reviewer #2

(Remarks to the Author)

The authors have addressed my concerns and clarified my confusion about the original version of the manuscript. The illustration models in Fig. 1b and their descriptions in the figure legend are helpful. The added Supplementary Table is also very helpful for readers in understanding the data sets used in the manuscript. The revised manuscript has been improved with additional information in the Instructions and figures. I believe it is now almost ready for publication.

Minor suggestion: Including a brief introductory information (perhaps in the Introduction) would help ensure that readers unfamiliar with ependymal cell development understand that at an early developmental stage (around Day 3), ependymal cells contain basal bodies at various stages of development, making them excellent samples for a structural study of basal body development.

Reviewer #3

(Remarks to the Author)

The authors have answered all comments and resolved issues raised by the reviewer sufficiently, further improving this excellent work. However, the reviewer still has one remaining concern considering the assignment of CEP41 to the density of mBMIP5. While the strategy to identify this density to be CEP41 is solid and supported by additional evidence, the EM data (shown in Figure 2 to the reviewer) of siRNA treated cells does not seem to further support the claims made in the corresponding section of the text. The explanation of the authors is reasonable. Still, the reviewer would recommend a more critical discussion in the main text and a more conservative interpretation.

The reviewer is also wondering if it would be possible to plot back the averages from the two different classes presented in Fig. 5c and quantify the average distance of the particles from the TZ. The distribution of the back-plotted particles could then be compared with the fluorescence data in Fig 6d. This could further support the conclusions drawn about the identification of CEP41 and its role in ciliogenesis.

Once this last concern is resolved, the reviewer would strongly recommend this manuscript for publication in Nature Communications.

Version 2:

Reviewer comments:

Reviewer #1

(Remarks to the Author)

I am satisfied with the revisions. In my opinion, the paper is now ready for publication.

Reviewer #3

(Remarks to the Author)

The detailed response of the authors resolves all my remaining questions and comments. It is perfectly reasonable to leave a more detailed analysis of the function of CEP41/mBMIP5 for future studies. The edits made to the discussion add clarity. The detailed explanation and analysis by plotting back the particles further substantiate the claims made about CEP41 and are much appreciated. Overall, the manuscript, already being impressive from the beginning, has significantly improved over the review process and I would recommend it for publication.

Minor comments:

L479: "However, it remains to be fully characterized of the exact roles and functional disparities of CEP41 between BBs and cilia"

The sentence could benefit from rephrasing.

REVIEWER COMMENTS

Reviewer #1 (Remarks to the Author):

Ma and coworkers provide a gorgeous example of how start-of-the-art structural techniques can resolve architectural features of biological machinery. By creating lamella from ciliated mouse ependymal cells (mEPCs) expressing GFP-labeled Deup1, they are able to use cryo-ET to visualize deuterosomes, basal bodies, and the proximal regions of axonemes. Using subtomogram averaging, they resolve subnanometer resolution reconstructions of the triplet microtubules (TMTs) of the basal body and the doublet microtubules (DMTs) of the transition zone and axoneme. These reconstructions improve on previous ex-vivo structures of bovine basal body-axoneme complexes but fall short of being able to identify most of the microtubule-bound proteins. Regardless of this limitation, the paper represents an impressive advance that should be published once the following issues have been addressed:

Major comments

1. *Biogenesis of basal bodies and cilia may vary in different organisms/cell types; the title should therefore reference that this study has been conducted in murine ependymal cells*

We thank the reviewer for their insightful comment and we have revised the title.

2. *The paper needs a table summarizing the different reconstructions; their origin (e.g. day-3 or day-10 mEPCs), their applied periodicity, particle number, and resolution. Currently, this information is spread throughout the text.*

We appreciate the reviewer's insightful comment. The relevant information has been summarized in **Supplementary Table S1**.

3. *In bovine samples, the DMT TZ is clearly demarcated with shelf-like MIPs in the A and B tubules (Greenan et al, 2019 JCB). These appear to be absent from the reconstructions presented here. Does this represent a difference between species, or a result of imaging “young” axonemes? Without this internal density, is the DMT TZ a distinct type of DMT, or simply a gradual transition between TMTs and DMTs? Exactly what is being imaged, and how this might relate to other species should be addressed.*

We appreciate the reviewer's insightful comment. We believe that the inconsistency of shelf-like MIPs in bovine and mouse TZ is caused by a difference between tissues. We acknowledged that shelf-like MIPs occupy the lumen of A and B tubules in bovine respiratory ciliary TZ. This shelf-like structure is not only present in the B tubule of human respiratory DMTs and the sperm DMTs of sea urchin, bull and mouse, but also in the sperm singlet microtubules at the flagellar endpiece and the proximal centriole of pig sperm¹⁻⁴. Recently, the molecules that constitute the shelf-like structure were identified as SPACA9 protein by high-resolution structures^{1, 4}. However, based on our previous work⁵, the MIPs are dramatically reduced in mature mouse ependymal cilia and the shelf-like structure is absent compared with that of bovine respiratory

cilia and mouse sperm flagella. On the other hand, our sub-tomogram averages contain basal bodies from different stages. We have performed detailed 3D classification which would have resolved the shelf-like structure if they are present. However, we did not see any classes containing the shelf-like structure, suggesting that its absence is an inherent nature of mouse ependymal transition zone, representing the tissue specificity rather than a difference of the structures in different ciliogenesis stages.

Our structural analyses showed that the mouse ependymal TZ possesses densities found in the basal body including mBMIP1-mBMIP7 (**Fig. 3**), as well as densities resembling the ciliary DMT such as CFAP20 at the inner junction. In addition, immunofluorescence staining of ac-tubulin showed continuous post-translational modification of basal-body, transition zone and axonemal (**Fig. 6d**). These results collectively indicate that TZ is a gradual transition between TMTs and DMTs.

4. The presence of densities consistent with axonemal MIPs (e.g. MNS1, CFAP53, NME7, CFAP141, and ENKUR) in the “DMT TZ” map suggests that 8 nm periodicity is being incorrectly applied and the true periodicity is higher.

We thank the reviewer for the insightful remarks. MIPs in doublet microtubules of cilia, transition zone or basal body exhibit various periodicities that are multiples of 8 nm such as 8 nm, 16 nm, 48 nm or even 96 nm in the case of *Tetrahymena*⁶. In practice, 8-nm maps can reveal finer structures of 8-nm densities because significantly more particles can be used for refinement, although higher periodicity densities are averaged out. Thus, 8-nm density maps are valuable. What’s more, as described in the “Particle picking and subtomogram averaging” section of the **Methods and Extended Data Fig 1i**, the 3D classification on all DMTs particles was applied with a cylinder mask near protofilaments B09-B10 (focused on the 16-nm-repeat CFAP52), resulting in two classes with 8-nm and 16-nm periodicities, respectively. The 8-nm-repeat particles are predominantly distributed in the TZ when we back-projected their coordinates in the raw tomograms. In addition, 3D classification on the particles in TZ, with cylinder masks near protofilaments A09-A10 (focused on the 48-nm-repeat CFAP52 and NME7), exclusively yielded the 8-nm periodicity. As illustrated in **Extended Data Fig. 7**, the 2D projections near protofilaments A09-A10 in the raw tomograms also revealed an 8-nm-repeat density. These results suggest that many regions that have higher periodicity in axonemal DMT exhibit 8-nm periodicity in the DMT of the transition zone. Although DMT^{TZ} likely exhibits an 8-nm-repeat based on our 3D classification focused on bulky globular features, we cannot rule out the possibility that higher periodicity exists for filamentous MIPs, such as MNS1 and CFAP53, given the limited particle number and resolution. Accordingly, we have refined the **Discussion** section to address this point.

5. More information should be provided on the deuterosomes. The central hub (CH) is a prominent feature of the tomograms but is not discussed in the paper. Is a reconstruction possible?

We thank the reviewer for the kind suggestions. For the deuterosomes, we have included

additional information and citations on deuterosomes in the **Introduction** of the revised manuscript as requested. For both the central hub and the cartwheel structures, tomograms clearly revealed the corresponding densities. However, due to the limited number of particles and their intrinsic flexibility, we are unable to obtain reliable average structures. The current results and the methods used to obtain them are presented in **Figure 1 to reviewer**.

Figure 1 to reviewer. The architecture of the central hub and cartwheel

A, A representative tomographic slice (26.6 nm) shows a basal body with densities corresponding to the central hub and cartwheel.

B-C, Subtomogram averaging results of the central hub and cartwheel. 182 particles along the central hub were picked a space of 25 nm and then cropped with a box size of 112 pixels and a pixel size of 13.31 Å/pix from 32 tomograms. The C9 symmetry was applied in the 3D refinement to obtain the averaged structure of the central hub. The limited data size and flexibility of the structures restricted the resolution of structural details.

6. Identifying proteins at 8 Å is challenging, although the identification of CEP41 appears convincing and is supported by additional evidence. A reconstruction of MTs from the CEP41 knockdown cells demonstrating loss of B tubules or CEP41 density would strongly support the conclusion. If this is not possible, tomography of the cells following knockdown should be done to show the gross morphological changes.

We thank the reviewer for pointing this out. To assess potential gross morphological changes, we performed cryo-ET and transmission electron microscopy (TEM) on samples indicated in **Figure 2 to reviewer**. While the overall morphology of basal bodies and axonemes in *CEP41* siRNA-treated mEPCs appeared normal, we noted a lower probability of capturing these structures in the knockdown samples compared to controls, consistent with our immunostaining results (**Fig. 6i-j**). Due to the limited dataset size of tomograms capturing ciliary axonemes in *CEP41* siRNA-treated mEPCs, we were unable to perform subtomogram averaging to investigate potential higher-resolution differences.

We attribute these negative results to the knockdown, rather than knockout, of *CEP41*. As shown in immunoblotting results (**Fig. 6h; Figure 2B to reviewer**), low levels of CEP41 were present in the cells treated with CEP41 siRNAs. Consistently, Cep41 immunofluorescence was detectable in basal bodies of these cells (**Fig. 6i, Figure 2F to reviewer**). We thus speculate that the basal bodies and axonemes observed in the *CEP41* RNAi samples were those whose formations were supported by the remaining CEP41.

Figure 2 to reviewer. Effect of *CEP41* knockdown on the cilia morphology and microtubule glutamylation in mEPCs

A, Experimental scheme. Serum starvation was performed at day 0 to induce multiciliation.

B, *CEP41* knockdown did not completely abolish its expression and reduce total levels of glutamylated tubulin.

C, 19.4-nm-thick cryo-electron tomographic slices show the normal ciliary axoneme (left) and

basal body (right) in *Cep41* siRNA-treated cells. Representative DMT (framed) and TMT (framed) are presented in insets of 38.8-nm-thick tomographic slices, respectively.

D-E, TEM images of ciliary axonemes (**D**) and basal body (**E**) reveal that *CEP41* knockdown in mEPCs did not trigger significant morphological changes in cilia. Representative cross-sections of ciliary axonemes (**D**, framed) and basal bodies (**E**, framed) are presented in insets. For mock-treated cells, 9 cross-sections of ciliary axonemes and 12 of basal bodies were analyzed. For *Cep41* siRNA-treated cells, 46 cross-sections of ciliary axonemes and 24 of basal bodies were analyzed.

F, Ciliary microtubule glutamylation in *Cep41* siRNA-treated cells was not impaired. mEPCs were immunostained for CEP41 and GT335 and counterstained with Odf2 to visualize basal bodies. Representative regions (framed) are presented in insets to show immunofluorescent images of multicilia (arrowheads) and primary cilia (arrows), respectively. In the inset showing control (NC) siRNA-treated multicilia, the two arrowheads point to immunofluorescent signals of CEP41 in basal bodies and cilia, respectively. In the inset showing *CEP41* siRNA-treated multicilia, a representative cilium (the one close to the arrowhead) was magnified by 2-fold to show details.

7. To test the model that *CEP41* enhances microtubule glutamylation, the authors should test whether tubulin glutamylation is reduced in the *CEP41* knockdown mEPCs.

Following the request, we examined tubulin glutamylation using the GT335 antibody. To our surprise, we did not observe a correlation between tubulin glutamylation and CEP41 (**Figure 2B to reviewer**). Firstly, immunoblotting reveals comparable total levels of glutamylated tubulin in mEPCs treated with control siRNA or *CEP41* siRNAs (**Figure 2B to reviewer**). Secondly, immunostaining of the same set of samples indicates that multicilia in *Cep41* siRNA-treated mEPCs were strongly positive for GT335, with mean immunofluorescent intensities comparable to those in control multicilia (**Figure 2F to reviewer**). GT335-decorated axonemes in *Cep41* siRNA-treated mEPCs were also generally short comparing with those in control cells (**Figure 2F to reviewer**), consistent with the results presented in **Fig. 6i-j** and **Extended Data Fig. 6d**. These multicilia, however, were negative for CEP41, though their basal bodies were positive for CEP41 (**Figure 2F to reviewer**). Similar results were observed in primary cilia (**Figure 2F to reviewer**). Nevertheless, as primary cilia are preexisted before serum starvation⁷, their tubulin glutamination may already be achieved before CEP41 levels were downregulated by the RNAi.

Our negative results in tubulin glutamylation may be attributed to incomplete depletion of CEP41 or a specific effect in multicilia. After all, the complete depletion of ciliary glutamylation has only been demonstrated by immunostaining primary cilia in fibroblasts from human patients carrying a *CEP41* splicing mutation that abolishes CEP41 expression⁸. Alternatively, our results may also raise concerns on the reported stimulation effect of CEP41 on tubulin glutamylation^{8,9}. Tubulin is glutamylated on its C-terminal tail, which is positioned outside the microtubule lamina, and the glutamylation occurs solely on the B09 microfilament^{10,11}. If TTLL6 is recruited into the lumen of the B tubule by CEP41 at B02, it is unlikely able to reach the C-terminal tails at B09 to glutamylate them. Furthermore, *CEP41* knockdown in mEPCs did not result in deformed A tubule (**Figure 2C-E to reviewer**) as observed in *cep41*

zebrafish morphants⁸. Due to these complexities, we choose not to present the results in **Figure 2 to reviewer** in the revised manuscript. We have modified the **Discussion** accordingly instead. We also hope our reviewer would agree that detailed functions of CEP41 is beyond the scope of the current work and can be addressed in future investigations.

8. The authors compare reconstructions of DMTs from day-3 and day-10 mEPCs to conclude that two MIPs are lost during maturation: mBMIP3L and mBMIP5. However, it is unclear if the reconstructions come from the same region of the axoneme. Are these differences a consequence of different stages of biogenesis or reconstructing more distal DMTs from the later time point?

We thank the reviewer for pointing this out. These differences are more likely a consequence of different stages of cilia biogenesis. For day-3 mEPCs, as described in the **Methods and Extended Data Fig. 1i**, we unbiasedly picked all the DMTs (including the TZ) for classification and averaging. We firstly separated particles from the TZ and axonemal DMTs. Further classification focused on mBMIP5 generated two classes of axonemal DMTs as shown in **Fig. 5c**. These two distinct classes of DMTs from day-3 mEPCs differ at mBMIP3 and mBMIP5. The day-10 results are from our previous work on the mature axonemal DMTs⁵, tomograms with undistorted “9+2” cilia ultrastructure were used for processing to avoid including any premature or very distal regions. Therefore, the new class found in the current work is likely unique to premature axoneme from day-3 mEPCs. We have revised the text and **Extended Data Fig. 1i**, and added EMDDB accession number of the day-10 DMT map in **Fig. 5c** for clarity.

9. Were TMT structures also compared from day-3 and day-10 mEPCs to show that they have the same MIP/MAP composition? This information should be included.

We thank the reviewer for pointing this out. Collecting a similar dataset for day-10 mEPCs is a significant undertaking that would require substantial time and effort. It's important to note that even within the day-3 mEPCs analyzed in this study, basal bodies and cilia exhibit various maturation stages. By employing extensive classification techniques, TMTs with different MIP/MAP compositions can be separated *in silico*, as demonstrated for DMTs in **Fig. 5c**. Therefore, it is unlikely that the MIP/MAP composition of TMTs undergoes drastic changes during maturation.

10. Both day-3 and day-10 can be considered “early” in the life cycle of cilia. The study would be enhanced by the addition of a much later time point. Does TMT/DMT structure change further?

We thank the reviewer for the suggestion. Cilia in mEPCs achieve their full size and typical back-and-forth beat pattern by day 10 (**Fig. 5d**)¹²⁻¹⁵. Importantly, when proper external directional flows are applied to the cultured mEPCs from day 5, their cilia can achieve planar cell polarity (PCP), i.e., cilia in different mEPCs beating towards a unified direction, by day 10¹². In the ependyma of mouse brains, the ciliary PCP is naturally achieved after P21 (21 days after birth)¹². These results strongly suggest that ependymal motile cilia are structurally mature

by day 10. Although we cannot absolutely rule out the possibility that TMT/DMT structures would change further after longer culture, considering the amount work required to achieve a solid conclusion on this, we hope our reviewer would agree that such an exploration is not necessary to our current manuscript and can be left to the future.

11. Movies should be provided showing the quality of the density and, in the case of CEP41, the quality of the fit to density.

We thank the reviewer for the suggestion and we have included **Supplementary Video S6**.

Minor comments

- *It should be made clear that the highest reported resolution (7.6 Å) is not biologically meaningful as it corresponds to a mixture of TMT identities and is only for the A and B tubules.*

We appreciate the reviewer's insightful comment and we have revised the information by adding the **Supplementary Table S1**.

- *The authors' statement that all BB MIPs/MAPs have 8-nm periodicity should be qualified because smaller densities which may have higher periodicities cannot be resolved at 8-9 Å resolution.*

We thank the reviewer for pointing this out and we have revised the **Discussion** section to address this point. "However, due to the limited subtomograms and resolution, we cannot definitively exclude the possibility that other small MIPs in BB and TZ may exhibit higher periodicities to support the periodic changes."

- *The authors suggest that they observe "strands of membrane particles that appear to anchor the ends of Y links" but also mention that they do not observe Y links. This inconsistency should be explained. Why are Y links not observed? Are proteins thought to be present in Y links expressed in mouse ependymal cells?*

We thank the reviewer for pointing this out. Y-links, a key element of the TZ, are hypothesized to enhance the mechanical stability of the TZ. It forms the bead-like ciliary necklace on the ciliary membrane, which we call "strands of membrane particles" in the text. MKS and NPHP modules compose the Y-links¹⁶, and CEP290/NPHP6, in particular, contains numerous coiled-coil domains. The Y-links are highly conserved among ciliated organisms and the ciliary necklace has been observed in mouse photoreceptors and respiratory cilia^{17, 18}. Therefore, the complexity and flexibility of Y-links, rather than the absence of proteins, make them challenging to observe and resolve using cryo-ET, as in previous studies of bovine tracheae, mouse retina, and *Chlamydomonas*¹⁹⁻²¹. To address the reviewer's concern, we have revised the text to "In addition, despite that the flexibility of Y-links make them challenging to observe in our tomograms, the ciliary necklace, strands of membrane particles that appear to anchor the ends of Y-links, was revealed at the TZ region".

- *The current results section contains speculation that is better suited to the discussion, e.g. “striations may contribute to the stiffness and compressibility of the TZ axoneme”.*

We thank the reviewer for pointing this out and we have revised the text.

- *The authors state that both intra- and extracellular pathways are required for efficient multiciliogenesis but do not show this experimentally. Without knowledge of the referenced papers, it is unclear how their observations revise current models of multiciliogenesis. Further explanation of the current models would be helpful to understand the implications of the new observations.*

We thank the reviewer for their kind suggestion. We have included more information in **Introduction**. In addition, we have performed the statistics on the extracellular and intracellular pathways in our data. Considering the thickness limitation (<200 nm) of the tomograms, the samples were restricted to tomograms where the ciliary sheath in the cytoplasm or the plasma membrane could be clearly distinguished. In our analysis of 261 tomograms, 38 tomograms presented cilia docked to the apical membrane (extracellular pathway), while 26 tomograms exhibited cilia either surrounded by vesicles or capped by the ciliary sheath in the cytoplasm (intracellular pathway).

- *Even in day-3 mEPCs, long cilia exist. It is possible to determine from the lamella the cilia length of the BBs being imaged?*

To have better contrast for subtomogram averaging, the thickness of lamella are less than 200 nm. Therefore, most cilia are truncated and it is impossible to estimate the accurate length of cilia imaged. For the BBs, because of their relatively short length, we were able to measure the length in some cases. Whenever such information is available, we documented this and the results are shown in **Extended Data Fig. 2b**.

- *PMID: 38841887 must be cited as it provides supporting evidence that CEP41 is a microtubule-binding protein.*

Thanks for pointing this out. We have included this reference.

- *A C2CD3 knockdown, followed by tomography to determine if rods were lost would enhance the study, but is not essential for the revision.*

We thank the reviewer for pointing out. We are still working on this, and so far the progress is slow because the knock down using siRNA is not efficient enough. We are designing other approaches to work out the identity of 27 rods.

- *Please provide additional information on how duplicated particles from oversampling were excluded, as duplicated particles can elevate resolution estimates.*

To exclude duplicated particles, we removed any particles that were within 4 nm of their nearest neighbors after alignment. We have included this information in the **Methods** session. It is also worth mentioning that if lots of duplicated particles exist in the final dataset and result in elevated resolution, RELION refinement should have a warning message and the FSC would not reach 0 at high frequency.

• Please clarify which particles were used to generate the 8.1 Å resolution map used to identify CEP41. Are these particles coming from different classes of microtubules?

We thank the reviewer for pointing out. We have revised the **Methods** section for clarity.

Reviewer #2 (Remarks to the Author)

In this manuscript, the authors utilized cryo-electron tomography to investigate the structures of ependymal cilium basal bodies and the microtubule doublets of nascent and mature ependymal cilia to clarify the assembly process of ependymal cilia. The specimens were prepared with cryo-FIB under the guidance of fluorescent imaging, which provided the authors with 177 useful tomograms for the structural analysis. The density maps after sub-volume analysis are of excellent quality, enabling the authors to determine the detailed structural differences of the basal bodies at the proximal, central, and distal regions and identify the doughnut-shaped densities as WD40 combining proteins, an 8-nm-repeat microtubule-associated protein in the lumen and a barrel of 27 rods at the border between the doublet and triplet microtubules. RNAi was also carried out in cooperation with cryo-ET, which enabled the determination of the identity of the mBMIP5 density to be CEP41. Overall, this manuscript represents a significant body of excellent experimental works that combine the high-resolution structural determination of the ependymal cilia basal body with detailed structural information of microtubule doublets in ependymal cilia at different maturation stages, as well as live cell imaging analysis of the ependymal cilia at various maturation stage. Besides providing detailed structural information on the ependymal basal body, the manuscript offers insight into the assembly process of microtubule doublets at the structural level. I think the research work in this manuscript is a very strong candidate for publication in Nature Communication.

We thank the reviewer for the positive assessment of our work.

I have the following concerns for the authors to consider:

Major concern:

The manuscript could profit from some reorganization, revision of statements, and additional background in the Introduction. In its current form, each part of the content is very good and relatively clearly written. However, these parts are not well-organized and linked, making it difficult for readers to appreciate the merit of the outstanding experimental works.

We thank the reviewer for the kind suggestion. In response to these suggestions, we have

refined certain statements and incorporated additional contextual information to strengthen the manuscript's presentation. Specifically, we have included additional information on deuterosome-dependent BB biogenesis in the **Introduction**, and provided a more detailed discrimination between the extracellular and intracellular pathways. Furthermore, we have added a supplementary figure summarizing the origin, periodicity and resolution of the structures in the study.

The manuscript's title emphasizes the “progressive biogenesis of basal bodies and cilia”. The first part of the Results describes the strategy of establishing a pipeline that captures various stages of ciliogenesis in mEPCs. Based on this section, Day 3, after serum starvation, provides multi-ciliated cells at various differentiation stages, including massive basal body biogenesis, while on Day 10, the majority shows long cilia. After this section, bearing the manuscript title in mind, readers would expect results of basal bodies of various stages (from I to IV? May be too much; At least several stages) to understand basal body biogenesis. However, the Results section only provided one complete structure of the basal body and its associated proteins. Indeed, the structural work of the basal body was carried out nicely, and the density map is excellent. It revealed structural differences at the basal body's proximal, central, and distal portions. Some microtubule-associated proteins, including the barrel of 27 sticks, were identified through convincing density map analysis. These are excellent structural information that represents a great deal of effort. However, there are six stages of the basal body genesis (Page 4, line 106, and Figure 1a and 1b); which stage of the basal body provided this cryo-ET map? The manuscript failed to provide this critical information in the context of basal body genesis. There is a significant disconnection of the logic thread for the whole manuscript. I am not requesting several basal body structures at different assembly stages since it may require significantly more work than practical. However, the current design and some statements in the manuscript lead to high expectations.

We thank the reviewer for the comments. We apologize for the lack of clarity in the previous manuscript. All the basal bodies and axonemes acquired in our tomograms were selected for subsequent cryo-ET analysis, regardless of their developmental stages. As illustrated in **Extended Data Fig. 2b**, the cryo-ET map represents an averaging structure of basal bodies with various stages.

Specifically, the process of massive BB biogenesis in mouse tracheal epithelial cells (mTECs) and mouse ependymal cells (mEPCs) has previously been divided into six stages (I to VI), based collectively on features of deuterosomes (size, number, and morphology) and BBs (number and relationship with deuterosomes) in a cell, revealed by immunofluorescence microscopy^{7, 22}. Generally, cells in stage I are defined by the presence of solely parental centrioles, in stage II by the presence of small deuterosomes with 0-2 procentrioles, in stage III by the presence of larger deuterosomes with more procentrioles, in stage IV by the emergence of Cep152-positive protrusions on deuterosomes, in stage V by the releasing of basal bodies from deuterosomes that are undergoing disassembly, and in stage VI by the formation of mature BBs that support ciliogenesis. Day-3 mEPCs encompass all six developmental stages⁷. Please note that although BBs also undergo progressive assembly from stage II to stage VI, the

assembly progresses of individual BBs on even the same deuterosome appear to vary (**Fig. 1e**). Therefore, we do not try to match the BB assembly stages defined by our cryo-ET studies (**Fig. 7**) with the stages of massive BB biogenesis.

To improve the clarity of the presentation, we have included illustration models for the typical cells in **Fig. 1b** and descriptions on features of cells at the indicated stages in the figure legend. We have also changed the BB assembly “stages” defined by our cryo-ET studies (**Fig. 7**) to “phases” to distinguish with the stages of massive BB biogenesis.

The basal body structure in the manuscript seems to be obtained with the mature basal bodies (Day 10). Let's assume so. How the single snapshot structural result, although an excellent structural snapshot, provides insight into progressive basal body genesis has yet to be established. The basis of the hypothesis proposed in Figure 7 is not clarified. It seems the manuscript assumes the structural difference from the proximal end to the distal end of the basal body represents the assembly process of the basal body. While it is possibly true, when no direct data backs up, there lacks a logical link between the structural data of the basal body and the basal body genesis model proposed in Figure 7. If so, the title of the manuscript is inappropriate, and some of the statements appear exaggerated, damaging the merit of this excellent research work. The manuscript should be revised with a better design. Suppose there is some cryo-ET data for the assembling basal bodies before they mature, as implied by a few inserts in Figure 7. In that case, the supporting data should be organized into a figure compared to the structural results of the mature basal body to support the hypothesis that the structural difference between the proximal and the distal region represents the different stages of the basal body assembly process.

We thank the reviewer for their insightful comment and apologize for the previous unclear description of sample preparation. As clarified above, all cryo-ET data were collected on Day 3 which possesses basal bodies at various stages of development as shown by immunofluorescence microscopy. For subtomogram averaging, the time resolution of different stages is potentially lost for basal body. On the contrary, in raw tomograms, we can readily distinguish basal bodies of different biogenesis stages, which is summarized in **Extended Data Fig. 2b**. For example, 27 rods are absent at the initiation stage and the central hub is gradually lost upon BB maturation. Moreover, we found that in most cases, the distal incomplete triplet microtubules (TMTs) are not present until BB reaches 250 nm, suggesting that the assemblies of complete and incomplete TMTs are sequential events during BB maturation. Our model in **Fig. 7** is supported by experimental data as shown in **Extended Data Fig. 2b**. We have revised the text to further support our conclusions. Additionally, we have deposited more tomograms in a public database for broader accessibility (**Figure 3 to reviewer and Methods**).

Figure 3 to reviewer. 2.7-nm-thick tomographic slices represent basal body biogenesis at various stages

(F) and (G) are the same tomographic slices of **Extended Data Fig. 4a-b** of the revised manuscript. BB, basal body; CH, central hub; Deu, Deuterosome; PM, plasma membrane.

The manuscript appears to be trying to capitalize on the research design for the structural investigation of ependymal ciliogenesis to strengthen its biological significance. While the microtubule doublet part fits the design, the basal body part, although excellent and containing abundant new information, is disconnected. The authors may consider focusing the cilia genesis study design on the microtubule doublet work and making a separate research justification for the basal body structural work. If so, Fig 7 is no longer essential. There is sufficient new structural information in the basal body structural work. This is not a request but a suggestion.

We thank the reviewer for their kind suggestion. As explained above, our results on the basal body represent multiple snapshots of its biogenesis, which we believe will provide valuable insights for researchers in this field.

The introduction part is too structure-focused. To capitalize on the biological insights obtained through the structural study in the manuscript, more background on ependymal cilia development, e.g., deuterosome-dependent multi-cilia genesis, should be added.

We thank the reviewer for their kind suggestion. We have revised the **Introduction** section to incorporate more detailed background information on deuterosome-dependent multi-cilia genesis.

Minor concern:

(1) Page 21 Line 524, Figure 3 legend: “mBMIP1 resembles MNS1 in cilia;” It is a confusing statement. How about “mBMIP1 resembles MNS1 in doublet microtubule;” ?

We thank the reviewer for the comment and we have revised the text.

(2) Figure 5. The overall summary of the figure states, “Nascent cilia revealed by in situ cryo-ET,” which is inappropriate. Panel 5d of the figure has nothing to do with the figure summary and it contains data for both nascent cilia on Day 3 and natural data from Day 10.

In fact, the structural comparison of the nascent (Day 3) and mature (Day 10) microtubule doublets represents a significant result in this manuscript, as it provides insights into the assembly process of microtubule doublets. The results in the Extended Information Figure 5, at least the S-5b, should be included in the main figures. For example, the S-5b may be included in current Figure 5. If so, the summary of Figure 5 may change to “Difference between the nascent and mature microtubule doublets”.

We thank the reviewer for the kind suggestions. We have revised the **Fig. 5** by including the Day-10 results for pairwise comparison and we have also changed the figure summary as suggested.

(3) The extended data Fig 3 demonstrates basal body template ciliogenesis, showing both extracellular and intracellular pathways. This is one of the key biological insights this manuscript provides. I suggest the author consider moving it to the main figures if space allows. Meanwhile, background information about extracellular and intracellular pathways of the basal body assembly regulation should be provided in the Introduction part.

We thank the reviewer for the comments and the background information about extracellular and intracellular pathways have been provided in the Introduction part (Line 54-Line 57). Considering that the main text already includes seven main figures and the need for further molecular-level research to fully elucidate these pathways, we keep the figure as an extended data figure.

(4) Page 35, Line 787, “ For immunoblotting, day-10 mEPCs (day-9 mEPCs for RNAi) were harvested. IMCD3 cells 787 were cultured for 2 days post serum starvation to stimulate the formation of primary cilia.” Did the authors introduce this part from other manuscripts but forget to modify?

We thank the reviewer for the comments, we have revised the text for clarify. Day-9 mEPCs were harvested to verify the knockdown efficiency of the CEP41 RNAi experiments (**Fig. 6h**). Day-10 mEPCs and day-2 IMCD3 cells were used to compare the CEP41 expression levels in motile and primary cilia (**Extended Data Fig. 6c**). Specifically, IMCD3 cells were cultured for 2 days post serum starvation to stimulate the formation of primary cilia.

Reviewer #3 (Remarks to the Author)

The authors of the manuscript “In situ cryo-electron tomography reveals the progressive biogenesis of basal bodies and cilia” establish a workflow to analyse structural details of ciliogenesis by employing a cryo-CLEM, cryo-FIB and cryo-ET workflow. To this end, myeloid endothelial progenitor cells (mEPCs) isolated from mouse brain were utilized as a model system for multiciliated cells and transduced with a Deup1-GFP fusion to enable targeted lamellae preparation. Cryo-ET data collection and subsequent subtomogram averaging allowed structural analysis of basal bodies (BB) in situ. Interestingly, tomograms revealed that mEPCs utilize both the intracellular and extracellular pathway for ciliogenesis, different from what was previously thought.

Based on the presence of certain structural features three different BB triplet microtubule (TMT) regions were defined for STA analysis and resolved to 9.1 Å (TMTPH), 8.8 Å (TMTIS_comC) and 9.2 Å (TMTIS_incomC). The subtomogram averages reveal common structural features of centriolar/BB TMTs like the pinhead and A-C linker, and variation in MIP decoration along the three different regions are observed. While the achieved resolution does not allow clear identification and assignment of MIPs and MAPs, several WD40 domains are identified to be closely associated with TMTs and speculated to be instrumental to enhance stability and integrity of BB TMTs. Based on STA combined with fitting of AF2 models the authors identify mBMIP5 to be CEP41. This is further supported by knock-down assays with CEP41-specific siRNA in mEPCs. STA of the transition zone doublet microtubules (DMTTZ) at a resolution of 12.1 Å showed a similar pattern of MIP decoration as the TMTIS_incomC region, however differences are observed at the inner junction. Furthermore, a distinct structure marking the transition from TMT to DMT, termed as rods, is described as a boundary to the TZ. Finally, the authors compile their results into a model that describes the different steps of ciliogenesis and highlight how MIP decoration evolves throughout the different regions.

The presented study provides some novel insight into ciliogenesis in situ. Although many of the structural features identified have been described before, the comprehensive nature of the study, covering the structural features along entire BB and providing an overview of these will be of significant value to the ciliogenesis field. Similarly, the observation that both extra- and intracellular ciliogenesis pathways appear in multiciliated cells advances insights into ciliogenesis pathways employed by different cell types. However, the reviewer would like to add that ciliogenesis is not their core expertise.

Since the resolution of the different STAs is not sufficient to clearly identify individual proteins, the significance of the structural insights on the BB TMT and DMT regions seems more limited. Yet, the study investigates these structures in a cell type previously not studied in that much detail and importantly from a metazoan, thus not only supporting previous STA studies but further extending our knowledge on how conserved these structures are.

The work largely supports the claims made. The data analysis and interpretation are mostly coherent. The conclusions drawn do sometimes seem to not be fully supported by the data. Thus, some clarification is required to ensure some of the data is not overinterpreted by the authors (see comments below).

The described methodology is sound and meets the expected standard within the field.

The material and methods are mostly described in enough detail to reproduce the work.

We thank the reviewer for the positive assessment of our work.

A few more details are required as indicated below.

Major

The reviewer is surprised that, while the central hub of the cartwheel is clearly visible in the tomograms, the spokes of the cartwheel do not seem to be present. As the spokes of the cartwheel are part of the final model (Figure 7), absence of the cartwheel spokes should at least be discussed.

We appreciate the reviewer's insightful comment regarding the cartwheel region. While the spokes of the cartwheel are indeed evident in our data, the inherent flexibility prevented us from obtaining a reliable, averaged structural representation. SAS-6, the critical component of the central hub and spokes, forms rod-shaped homodimers via interactions within their N-terminal domains. Nine homodimers assemble into the central hub emanating nine coiled-coil rods (known as spokes)²³. Additionally, from proximal to distal BBs, the twist angles between adjacent TMTs vary^{19, 24}, resulting in the longitudinal offsets of the spokes. The offsets, along with the coiled-coil structure of spokes, lead to a lower contrast in tomograms. Previous cryo-ET structures of spokes were obtained by resolving the structure of the cartwheel, and were of low-resolution^{25, 26}. Moreover, such density in mammalian samples is weaker than in other species analyzed. Here we show the averaged structure of cartwheel with nine spokes radiating outwards (**Figure 1 to reviewer**). To address this, we have revised the text and included a tomographic slice in the revised **Fig. 7**, which provides a more detailed visualization of the cartwheel region.

Figure 1 to reviewer. The architecture of the central hub and cartwheel

A, A representative tomographic slice (26.6 nm) shows a basal body with densities corresponding to the central hub and cartwheel.

B-C, Subtomogram averaging results of the central hub and cartwheel. 182 particles along the central hub were picked a space of 25 nm and then cropped a the box size of 112 pixels and a

pixel size of 13.31 Å/pix from 32 tomograms. The C9 symmetry was applied in the 3D refinement to obtain the averaged structure of the central hub. The limited data size and flexibility of the structures restricted the resolution of structural details.

The authors define TMTIS_incomC as a category of TMT before the transition zone. While previous studies and some of the raw data presented here (Extended data Figure 4 a, b and d) suggest a gradual change from a full C-tubule to a missing C-tubule, the STA presented in figure 2A suggests a clear class of 6 PF C-tubule in that region. It should be discussed in the main text how sharp this transition really is (meaning: is it really full C-tubule, 6PF C-tubule, no C-tubule).

We appreciate the reviewer for the valuable suggestion. The transition from TMT^{IS_comC} to TMT^{IS_incomC} can be attributed to the gradual loss of PFs, while the transition from TMT^{IS_incomC} to DMT seems a sharp transition. We have included a new **Extended Data Fig. 2c** to illustrate the process clearly. Therefore, TMT^{IS_incomC} represents an averaged structure of the transition, with the C tubule exhibiting a lower resolution compared to the A and B tubules (**Figure 4 to reviewer**). We have revised the text accordingly.

Figure 4 to reviewer. The cryo-EM map of TMT^{IS_incomC} colored by local resolution

The authors state the observation that BBs template ciliogenesis via extra- and intracellular pathways. Since the field previously expected multiciliated cells to prefer the extracellular pathway, it would be great to substantiate the authors claim with a quantitative comparison. The size of the dataset should allow some statistical analysis.

We thank the reviewer for their kind suggestion. we have performed the statistic and included this information in the revised manuscript. Considering the limitation of the thickness (<200 nm) of the tomograms, the samples were restricted to tomograms where the ciliary sheath in the cytoplasm (for intracellular pathway) or the plasma membrane (for extracellular pathway) could be clearly distinguished. In brief, in our analysis of 261 tomograms, 38 tomograms presented cilia docked to the apical membrane (extracellular pathway), while 26 tomograms exhibited cilia either surrounded by vesicles or capped by the ciliary sheath in the cytoplasm (intracellular pathway).

It is not clear to the reviewer, which data exactly (TMT or DMT, which part of TMT/DMT?) was used and refined to identify CEP41. This needs to be more clearly specified either in the corresponding methods section or in the main text and corresponding figure. Furthermore, at

the present resolution the reviewer would suggest a more conservative interpretation of the identification. The authors claim that based on cross-correlation CEP41 is the most probable candidate, but clear identification seems not fully supported by the average.

Figure 5 to reviewer. The fitting of the top 30 proteins from mouse proteosome into the CEP41 density

We thank the reviewer for the comment. Both the density maps of CEP41 in TMTs and DMTs were resolved individually, with CEP41 in TMT achieving a higher resolution of 8.1 Å. The map has been deposited in EMDB (EMD-39658) for CEP41 identification and model refinement. We have revised the description in the main text and added the detailed methods. Given the size of the CEP41 density, the AlphaFold2 models of 13,016 proteins below 45 kDa from the mouse proteome were firstly fitted into the density automatically by Colores program of the Situs package and repeated twice. CEP41 was ranked 27th and 28th with a cross-correlation score of 0.6164 in the two repeated experiments. We then manually fitted the top 30 proteins with the highest cross-correlation scores to the density in Chimera (**Figure 5 to reviewer**), and CEP41 emerged as the most likely candidate. To be note, the core of CEP41

AlphaFold2 model with high predicted local-distance difference test (pLDDT) values fits well with the cryo-EM density while the flexible regions with low pLDDT values stretch out of the cryo-EM density, suggesting a good fitting (**Extended Data Fig. 6a**). Other candidates, although they may have a higher cross-correlation score, are excluded based on the mismatch of PDB with the density map (**Figure 5 to reviewer**). Besides, the structural fitting result, the dynamic expression levels and distribution of CEP41 correspond with its presence in nascent cilia and absence in mature cilia. In addition, the axoneme tips, with the constant turnover of tubulins, were decorated with the CEP41 immunostaining, further supporting our assumption.

The siRNA experiment particularly quantifies cilia formation. It would be important to also quantify another baseline parameter, like viability, as a control and to show that CEP41 knock-down does not affect other important cellular functions.

We thank the reviewer for the comments. We analyzed images acquired in experiments for **Fig. 6g-j** and **Figure 2 to reviewer**. There were no significant differences in cell numbers between WT and *CEP41*-knockdown mEPCs (**Figure 6 to reviewer**), suggesting that the *CEP41* knockdown has minimal impact on cell proliferation.

Figure 6 to reviewer. *CEP41* knockdown does not impair cell proliferation

A, mEPCs were immunostained for DAPI and acetylated tubulin (Ac-tub) to visualize nucleus and cilia, respectively. The image was captured with a 40× objective measuring 290 μm × 290 μm. NC, negative control.

B, The circle or square data points represent the mean cell numbers in three images for each experiment. A total of five experiments were conducted (3 from day-9 mEPCs and 2 from day-4 mEPCs). The paired results were connected by lines. *P*-value was from two-sided student's *t*-test.

Line 304-310: In this paragraph, DMT structures from nascent and mature DMT48nm are compared. It is not completely clear to me whether nascent automatically means that the structures come from 3-day mEPCs and mature from 10-day mEPCs. Furthermore, it seems to me that this is the first time averaging of 10-day mEPCs appears and that in the material and methods only 3-day mEPCs are mentioned to be used for preparation of grids. Where is the average of the 10-day mEPCs coming from? Why is it only shown in the extended figure? To allow a reader an easy comparison, as described in the text, it would be nice to have a panel in the main figure. Also, the reviewer is wondering whether radial spokes, IDA or ODA were more clearly visible in averages from the mature 10-day mEPCs

We thank the reviewer for the comment and have revised the corresponding description in the

main text. The nascent DMT structure, presented in our paper come from day-3 mEPCs as clarified in **Fig. 1a** and **Methods**. The mature DMT^{48nm} structure (EMD-37111) from day-10 mEPCs has been published⁵ with a resolution of 19.6 Å. In addition, the DMT^{96nm} structure (EMD-37104) from day-10 mEPCs, reported in the same paper, indeed exhibited clearly visible radial spokes, IDA and ODA. To enhance the understanding of the structural differences between nascent and mature cilia, we have rearranged the figures in **Fig. 5** and **Extended Data Fig. 5**.

Minor

The density in cyan correlated to POC1B at IJAB (Figure 2a,b,c,d,f) seems to stretch in two directions, towards the A and the B tubule. However, when fitting POC1B a stretch to only one side is fitted with the extending α -helix of POC1B. Does the other extended density not belong to POC1B? If so, why is it the same colour?

We thank the reviewer for the insightful observation. The inner junction between A and B tubules appears to fulfill its architectural role from both sides, which justifies the uniform color coding in our representation. The color here represents a possible functional complex rather than a single protein. Based on previous studies²⁵, we hypothesize that the densities of IJ^{AB} are likely composed of both POC1B and a part of CEP135. Although POC1B consists of one WD domain and one α -helix, the resolution limitations of our data preclude a definitive assignment of its conformation. Therefore, the model fitting in **Fig. 2f** aligns with the segment with the highest likelihood of association, but we cannot rule out the contribution of the other extended density.

In a few instances, the manuscript would profit from more clarity in writing and a more coherent connection between the figures and different structural features highlighted in the text. A (non-exhaustive) list of examples and a few typos can be found below:

Line 86: “larger than” instead of “large than”?

We thank the reviewer for the comment and we have revised the text.

Line 125: “the structure” refers to the cartwheel?

We thank the reviewer for the comment and we have revised “the structure” to “the cartwheel”.

Line 256-260: First, periodicities and identities of MIPs of DMTTZ is said to be like TMT, except for inner junction. But in the next sentence the DMTTZ inner junction is compared to ciliary axoneme DMTs. What is the difference in the IJ between TMT and DMTTZ? Then afterwards it can still be compared to ciliary axoneme. The authors could also highlight this difference in the according figure (figure 3).

We thank the reviewer for the feedback. We have revised the sentence in the main text to avoid

any confusion. The inner junctions between TMTs and DMT^{TZ} are remarkably different, transitioning from the proposed POC1B and WDR90 in TMTs to the exclusive CFAP20 in DMT^{TZ}. The IJ of the DMT^{TZ} will progress to the IJ of the ciliary axoneme with the PACRG incorporation.

Line 270: “in another word” should be “in other words”

We thank the reviewer for the comment and we have revised the text.

Line 442-445: It is not clear to me what this sentence aims at. Rephrasing it would definitely help to make it clearer.

We thank the reviewer for the comment. The sentence is intended to emphasize the sequential transformation of the IJ components and the periodic variations from proximal BBs to distal axonemes. We have revised the text.

Figure 6: what does the cyan colour indicate? Or is the red colour just indicating the RHOD and the cyan the “not-RHOD”? It would be nice to indicate into which direction disordered loops would continue (if possible without making the figure visually incomprehensible).

We thank the reviewer for the comment. The red color indeed indicates the RHOD domain and cyan represents for the α -helices. We have made the necessary revisions to **Fig. 6b** and its figure legends. The disordered loop illustrated in **Extended Data Fig. 6a** was excluded from the **Fig. 6b,c** for its high flexibility and its poor-quality density.

Extended data Figure 4 a: sentence in line 974 and 975 is a bit unclear. The white box in the figure is hard to see.

We thank the reviewer for the comment. We have revised the figure and its figure legends.

References for reviewers:

1. Gui M, *et al.* SPACA9 is a luminal protein of human ciliary singlet and doublet microtubules. *Proceedings of the National Academy of Sciences of the United States of America* **119**, e2207605119 (2022).
2. Leung MR, *et al.* The multi-scale architecture of mammalian sperm flagella and implications for ciliary motility. *The EMBO journal* **40**, e107410 (2021).
3. Chen Z, *et al.* De novo protein identification in mammalian sperm using in situ cryoelectron tomography and AlphaFold2 docking. *Cell* **186**, 5041-5053 e5019 (2023).
4. Leung MR, *et al.* Structural specializations of the sperm tail. *Cell* **186**, 2880-2896 e2817 (2023).
5. Meng X, *et al.* Multi-scale structures of the mammalian radial spoke and divergence of axonemal complexes in ependymal cilia. *Nature communications* **15**, 362 (2024).
6. Kubo S, *et al.* Native doublet microtubules from *Tetrahymena thermophila* reveal the importance of outer junction proteins. *Nature communications* **14**, 2168 (2023).

7. Zhao H, *et al.* Parental centrioles are dispensable for deuterosome formation and function during basal body amplification. *EMBO reports* **20**, e46735 (2019).
8. Lee JE, *et al.* CEP41 is mutated in Joubert syndrome and is required for tubulin glutamylation at the cilium. *Nature genetics* **44**, 193-199 (2012).
9. Ki SM, *et al.* CEP41-mediated ciliary tubulin glutamylation drives angiogenesis through AURKA-dependent deciliation. *EMBO reports* **21**, e48290 (2020).
10. Yang WT, *et al.* The Emerging Roles of Axonemal Glutamylation in Regulation of Cilia Architecture and Functions. *Front Cell Dev Biol* **9**, 622302 (2021).
11. Alvarez Viar G, *et al.* Protofilament-specific nanopatterns of tubulin post-translational modifications regulate the mechanics of ciliary beating. *Curr Biol* **34**, 4464-4475 e4469 (2024).
12. Guirao B, *et al.* Coupling between hydrodynamic forces and planar cell polarity orients mammalian motile cilia. *Nat Cell Biol* **12**, 341-350 (2010).
13. Zhao H, *et al.* Fibrogranular materials function as organizers to ensure the fidelity of multiciliary assembly. *Nature communications* **12**, 1273 (2021).
14. Delgehyr N, *et al.* Chapter 2 - Ependymal cell differentiation, from monociliated to multiciliated cells. In: *Methods in Cell Biology* (eds Basto R, Marshall WF). Academic Press (2015).
15. Zheng J, *et al.* Microtubule-bundling protein Spef1 enables mammalian ciliary central apparatus formation. *J Mol Cell Biol* **11**, 67-77 (2019).
16. Williams CL, *et al.* MKS and NPHP modules cooperate to establish basal body/transition zone membrane associations and ciliary gate function during ciliogenesis. *J Cell Biol* **192**, 1023-1041 (2011).
17. Mercey O, Mukherjee S, Guichard P, Hamel V. The molecular architecture of the ciliary transition zones. *Curr Opin Cell Biol* **88**, 102361 (2024).
18. Park K, Leroux MR. Composition, organization and mechanisms of the transition zone, a gate for the cilium. *EMBO reports* **23**, e55420 (2022).
19. Greenan GA, Vale RD, Agard DA. Electron cryotomography of intact motile cilia defines the basal body to axoneme transition. *J Cell Biol* **219**, (2020).
20. van den Hoek H, *et al.* In situ architecture of the ciliary base reveals the stepwise assembly of intraflagellar transport trains. *Science* **377**, 543-548 (2022).
21. Zhang Z, Moye AR, He F, Chen M, Agosto MA, Wensel TG. Centriole and transition zone structures in photoreceptor cilia revealed by cryo-electron tomography. *Life Sci Alliance* **7**, e202302409 (2024).
22. Zhao H, *et al.* The Cep63 paralogue Deup1 enables massive de novo centriole biogenesis for vertebrate multiciliogenesis. *Nat Cell Biol* **15**, 1434-1444 (2013).
23. Kitagawa D, *et al.* Structural basis of the 9-fold symmetry of centrioles. *Cell* **144**, 364-375 (2011).
24. Le Guennec M, *et al.* A helical inner scaffold provides a structural basis for centriole cohesion. *Sci Adv* **6**, eaaz4137 (2020).
25. Klena N, *et al.* Architecture of the centriole cartwheel-containing region revealed by cryo-electron tomography. *The EMBO journal* **39**, e106246 (2020).
26. Guichard P, *et al.* Native architecture of the centriole proximal region reveals features underlying its 9-fold radial symmetry. *Curr Biol* **23**, 1620-1628 (2013).

1 **REVIEWER COMMENTS**

2
3 **Reviewer #1 (Remarks to the Author):**

4
5 *I thank the authors for their thoughtful responses to my comments and questions. I also*
6 *appreciate their efforts to investigate the relationship between CEP41 and tubulin*
7 *glutamylolation and agree that the preliminary results are beyond the scope of the paper and the*
8 *question is best addressed in future investigations. Overall, the changes and additions have*
9 *improved the paper, and I now recommend it unreservedly for publication.*

10
11 *We thank the reviewer for the positive assessment of our work and for his/her valuable inputs*
12 *that have helped us to substantially improve the manuscript.*

13
14 *Minor comments:*

15
16 *L84. The newly introduced sentence: “Deup1 (deuterosome protein 1, also known as Ccdc67),*
17 *a paralogue of Cep63, which is a mother centriole-localized, Cep152-interacting protein*
18 *important for centriole biogenesis, forms the core of deuterosomes” would benefit from*
19 *simplification.*

20
21 *We thank the reviewer for the kind suggestion. In response to this suggestion, we have revised*
22 *the text (L82-83) to “Deup1 (deuterosome protein 1), a paralogue of the mother centriole-*
23 *localized protein CEP63, forms the core of deuterosomes”.*

24
25 *L100. “High-resolution” and “entire” should be removed from this sentence.*

26
27 *We thank the reviewer for the comment and we have revised the text.*

28
29 *L170 “The protofilaments of C tubule in TMTIS_comC was gradually shedding from C10*
30 *during proximal to distal BBs (Extended Data Fig. 2c).” This sentence could be revised for*
31 *clarity.*

32
33 *We thank the reviewer for the comment. We have revised the sentence (L167-168) to “Along*
34 *the proximal-to-distal axis of BBs, the loss of C-tubule protofilaments initiates at C10 and*
35 *proceeds to C01 (Extended Data Fig. 2c)” , and modified the Extended Data Fig. 2c*
36 *accordingly.*

37
38 **Reviewer #2 (Remarks to the Author):**

39
40 *The authors have addressed my concerns and clarified my confusion about the original version*
41 *of the manuscript. The illustration models in Fig. 1b and their descriptions in the figure legend*
42 *are helpful. The added Supplementary Table is also very helpful for readers in understanding*
43 *the data sets used in the manuscript. The revised manuscript has been improved with additional*
44 *information in the Instructions and figures. I believe it is now almost ready for publication.*

45

46 *Minor suggestion: Including a brief introductory information (perhaps in the Introduction)*
47 *would help ensure that readers unfamiliar with ependymal cell development understand that at*
48 *an early developmental stage (around Day 3), ependymal cells contain basal bodies at various*
49 *stages of development, making them excellent samples for a structural study of basal body*
50 *development.*

51

52 We appreciate the reviewer's constructive feedback and would like to take this opportunity to
53 express our thankfulness. As per the suggestion, we have revised the manuscript (L106-111) to
54 clarify the rationale for selecting day-3 mEPCs. The revised text is "In the study, we used
55 mEPCs cultured to day 3 post serum starvation, a timepoint capturing multiciliated cells at
56 various differentiation stages (**Fig. 1a,b**), for structural analysis. This heterogeneity, where BBs
57 at distinct developmental phases coexist, makes the early-stage ependymal cells a powerful
58 model for capturing structural snapshots of BB assembly". In addition, we have refined the
59 **Introduction** section (L88-L90) of the revision manuscript to incorporate details on the
60 benefits of the EPC system. The revised text is "Ependymal cells, with longer cilia and larger
61 deuterosomes than those in tracheal epithelial cells, were chosen to investigate BB biogenesis
62 and multiciliogenesis in the study".

63

64 **Reviewer #3 (Remarks to the Author):**

65

66 *The authors have answered all comments and resolved issues raised by the reviewer sufficiently,*
67 *further improving this excellent work. However, the reviewer still has one remaining concern*
68 *considering the assignment of CEP41 to the density of mBMIP5. While the strategy to identify*
69 *this density to be CEP41 is solid and supported by additional evidence, the EM data (shown in*
70 *Figure 2 to the reviewer) of siRNA treated cells does not seem to further support the claims*
71 *made in the corresponding section of the text. The explanation of the authors is reasonable.*
72 *Still, the reviewer would recommend a more critical discussion in the main text and a more*
73 *conservative interpretation.*

74

75 We appreciate the reviewer's positive evaluation of our work. Our cryo-ET (**Fig. 5c**),
76 immunostaining (**Fig. 6d-f**), and motility (**Fig. 5d**) results consistently suggest that CEP41
77 binds to freshly assembled axonemes and is then dissociated following the maturation of the
78 axonemal region (**Fig. 7**). The RNAi experiments further support a role of CEP41 in ciliary
79 formation (**Fig. 6g-j and Extended Data Fig. 6b-d**). While we acknowledge that the present
80 study does not fully elucidate the detailed functions of CEP41/mBMIP5 due to technical
81 limitations, we hope our reviewer would agree that comprehensive functional characterizations
82 lie beyond the scope of this current investigation and are more appropriate for future researches.
83 In accordance with the reviewer's recommendation, we have revised the manuscript (L344,
84 L381, L388-390, L479-480 and L488-490) to incorporate a more critical discussion and a
85 correspondingly conservative interpretation of our findings.

86

87 *The reviewer is also wondering if it would be possible to plot back the averages from the two*
88 *different classes presented in Fig. 5c and quantify the average distance of the particles from the*

89 *TZ. The distribution of the back-plotted particles could then be compared with the fluorescence*
90 *data in Fig 6d. This could further support the conclusions drawn about the identification of*
91 *CEP41 and its role in ciliogenesis.*

92

93 We appreciate the reviewer's insightful suggestion. While the idea is compelling, implementing
94 it presents several challenges. The limited field of view makes simultaneously capturing both
95 the TZ and the tip of mature cilia impossible. In addition, day-3 mEPCs we used are in very
96 early stages of development, as evidenced by a higher ratio of CEP41-containing to CEP41-
97 lacking DMT particles (81:19).

98

99 Despite these constraints, we performed the plotting back analysis as recommended by the
100 reviewer (**Figure 1 to reviewer**). For nascent cilia with visible basal bodies and tips, all
101 DMT^{48nm} particles contained CEP41, corroborating our hypothesis (**Figure 1A-D to reviewer**).
102 In longer cilia, while CEP41-lacking particles were sufficiently observed and locally enriched,
103 CEP41-containing particles were also maintained at both proximal and distal cilia segments
104 (**Figure 1E-I to reviewer**). Although the limited ciliary length resolved in tomograms hindered
105 robust statistical assessment of particle distribution, these results are consistent with the
106 immunofluorescence results that CEP41 localized along the axoneme of nascent cilia and at the
107 tip region of long cilia (**Fig. 6d-f**), possibly due to the assembly of nascent DMTs in the tip
108 region.

109

110 **Figure 1 to reviewer. Plotting back analysis of CEP41-containing and CEP41-lacking**
 111 **DMT^{48nm} particles in cilia**

112 (A, C) 2.7-nm-thick tomographic slices show the nascent ciliogenesis via the intracellular (A)
 113 or extracellular (C) pathway. The BB, TZ and cilia region are separated by the dashed yellow
 114 lines. The proximal/minus (-) and distal/plus (+) ends of cilia are indicated.

115 (B, D) Plotting back analysis of DMT^{48nm} particles corresponding to the cilia in (A) and (C).
 116 The cilia are completely assembled from CEP41-containing DMT^{48nm} particles.

117 (E-H) 2.7-nm-thick tomographic slice (E) and the corresponding DMT^{48nm} particles (F-H)
 118 mapped from three individual ciliary segments. The proximal/minus (-) and distal/plus (+)
 119 ends of cilia are indicated. Green spheres denote CEP41-containing DMT^{48nm} particles and red
 120 spheres represent CEP41-lacking DMT^{48nm} particles (F-H). The TZ region was cropped by the
 121 FIB milling and is outside the tomogram.

122 (I) Distance analysis of CEP41-containing and CEP41-lacking DMT^{48nm} particles relative to
 123 the proximal segments of three cilia (F-H), respectively. DMT^{48nm} particles were numbered at
 124 15 nm intervals with the ciliary proximal end within the tomogram (E) as a starting point.

125 BB, basal body; TZ, transition zone; PM, plasma membrane; CM, ciliary membrane.

126

127 *Once this last concern is resolved, the reviewer would strongly recommend this manuscript for*
128 *publication in Nature Communications.*

129

130 *With the revisions and clarifications, we hope our reviewer would agree that the manuscript is*
131 *now appropriate for publication. We sincerely thank the reviewer for helping us to improve the*
132 *manuscript.*

REVIEWER COMMENTS

Reviewer #3 (Remarks to the Author):

The detailed response of the authors resolves all my remaining questions and comments. It is perfectly reasonable to leave a more detailed analysis of the function of CEP41/mBMIP5 for future studies. The edits made to the discussion add clarity. The detailed explanation and analysis by plotting back the particles further substantiate the claims made about CEP41 and are much appreciated. Overall, the manuscript, already being impressive from the beginning, has significantly improved over the review process and I would recommend it for publication.

Minor comments:

L479: “However, it remains to be fully characterized of the exact roles and functional disparities of CEP41 between BBs and cilia” The sentence could benefit from rephrasing.

We thank the reviewer for the kind suggestion. We have revised the text to “However, the exact roles and functional disparities of CEP41 between BBs and cilia remain to be fully characterized”.